# Dynamic birefringence and chirality of magnetically controllable assemblies of anisotropic plasmonic nanoparticles in dispersion

Hyojung Kang [1], Yoojung Jeon [1], Kyungnae Baek [1], SeonJu Park [1], Jayoon Lee [1], Tae Soup Shim [2,3], Jerome K. Hyun [1] ✉ & So-Jung Park [1,4] ✉

The ability to control the orientation and arrangement of plasmonic nanoparticles with shape anisotropy offers a promising route to achieving highly tunable optical properties. In this study, we introduce a synthetic approach for magnetically controllable plasmonic nanoparticles (MPs) consisting of an anisotropic gold core encapsulated by an iron oxide shell. The superparamagnetic property of the iron oxide shell enables rapid, reversible, and remotely controlled alignment of MPs, allowing for dynamic manipulation of their optical properties. Linearly aligned MPs demonstrate tunable transmission colors via plasmon-mediated birefringence. Helical MP arrays exhibit circular dichroism of up to 12° and g-factors reaching 0.21—the highest reported value for solution-phase assemblies of achiral nanoparticles. The synthetic method is applicable to nanoparticles of various sizes and shapes, highlighting its generality and expandability.

Metal nanoparticles (NPs) have garnered significant attention due to their size- and shape-dependent optical properties originating from the localized surface plasmon resonance (LSPR)[1–3]. Metal NPs with anisotropic shapes are of particular interest because they possess multiple plasmonic modes, which can be selectively excited depending on the orientation of NPs relative to the polarization of the incident light. For instance, gold nanorods (AuNRs) exhibit transverse LSPR (T-LSPR) and longitudinal LSPR (L-LSPR) modes, which correspond to the electron oscillation along their short and long axes, respectively. Various methodologies have been adopted to attain orientation-controlled AuNR arrays, which include solvent evaporation[4,5], mechanical brushing[6–8], templating[9], and the application of an electric or magnetic field[10,11]. Among them, the magnetically induced alignment of plasmonic NPs is particularly intriguing because it provides a means of dynamically and remotely controlling NP alignment with rapid responsivity and complete reversibility[12].

Several different approaches have been reported for synthesizing magnetically controllable plasmonic nanoparticles (MPs)[13–21]. For example, Yin et al. developed a space-confined seeded growth method to synthesize Au/Fe$_x$O$_y$ hybrid nanorods[13–15]. This method, which utilizes FeOOH nanorods as a template, created a void space along the FeOOH nanorod for AuNR growth via subsequent silica and polymer coating, FeOOH reduction, silica etching, and surface modification to finally generate Au/Fe$_3$O$_4$ nanorods embedded in a polymer shell. Lee et al. have reported a simple one-pot solvothermal method for synthesizing Au/Fe$_x$O$_y$ core/shell nanowires with high aspect ratios[16,17]. Park and coworkers synthesized multiblock nanorods by electrochemically depositing magnetic and plasmonic segments in anodized aluminum oxide channels[18,19]. However, these methods offered limited control over the size and shape of MPs, being constrained by the shape of the template or the inherent characteristics of the synthetic method.

[1]Department of Chemistry and Nanoscience, Ewha Womans University, Seoul, Republic of Korea. [2]Department of Energy Systems Research, Ajou University, Suwon, Republic of Korea. [3]Department of Chemical Engineering, Ajou University, Suwon, Republic of Korea. [4]Graduate Program in Innovative Biomaterials Convergence, Ewha Womans University, Seoul, Republic of Korea. ✉e-mail: kadam.hyun@ewha.ac.kr; sojungpark@ewha.ac.kr

Here, we developed a synthetic approach for MPs composed of an AuNP core and an iron oxide ($Fe_xO_y$) shell; the superparamagnetic $Fe_xO_y$ shell allows for rapid magnetic control of the NP orientation, while the AuNP core provides tunable LSPR properties. Our synthetic method uses AuNPs pre-synthesized via well-established procedures as the core material. Therefore, the size, shape, and LSPR characteristics of the NPs can be selected with a high degree of freedom. Compared to existing methods, this approach is particularly amenable to producing MPs with smaller sizes, possessing high stability, magnetic controllability, and useful optical properties. Importantly, the MP arrays formed under a magnetic field exhibited intriguing polarization-dependent optical properties beyond the simple color transition due to selective LSPR excitation. We demonstrate that MPs aligned along a linear magnetic field exhibit strong plasmon-mediated birefringence, which is manifested by a wide range of transmitted colors. Furthermore, the MPs under a helical magnetic field demonstrated giant chirality with a *g*-factor of 0.21, which is among the highest values reported thus far for plasmonic assemblies. We believe that this work will find application in real-time control of optical properties in areas such as information storage[22], displays[23], smart windows[24], and chiral sensing[25].

## Results

### Synthesis and characterization of MPs

Figure 1a illustrates our synthetic approach for integrating plasmonic and magnetic properties into MPs comprising a gold core and an $Fe_xO_y$ shell. AuNRs were chosen as the core material because of their shape anisotropy and tunable optical properties. Figure 1b

presents the transmission electron microscopy (TEM) image of our prototype AuNRs with an aspect ratio of 2.3 (34 ± 3 nm by 81 ± 6 nm). Prior to $Fe_xO_y$ coating, the hexadecyltrimethylammonium bromide (CTAB) medium of the AuNR solution was exchanged with a hexadecyltrimethylammonium chloride (CTAC) solution to facilitate further reactions. Then, $K_2PtCl_4$ and $FeCl_2 \cdot 4H_2O$ were sequentially added to the AuNR solution, and the mixture was heated at 100 °C for 1 h. The redox reaction between $PtCl_4^{2-}$ and $Fe^{2+}$ generates an initial iron oxyhydroxide (FeOOH) coating on AuNRs, upon which further deposition of FeOOH occurs through the oxidation of $Fe^{2+}$ by dissolved oxygen[26] (Supplementary Note 1). Our optimized experimental condition (0.4 mM of $K_2PtCl_4$, 10 mM of $FeCl_2 \cdot 4H_2O$, and 4.4 nM of AuNRs) resulted in a uniform FeOOH shell with a thickness of 14 ± 1 nm (Fig. 1c; Supplementary Fig. 1). The small amount of $K_2PtCl_4$ was required to form a uniform FeOOH shell (Supplementary Fig. 2), and the shell thickness and morphology were controlled by adjusting the ratio of AuNRs to iron precursors (Supplementary Figs. 3 and 4). Without the CTAB-to-CTAC surfactant exchange step mentioned above, the FeOOH shell formation was significantly suppressed, as shown by TEM and extinction spectra (Supplementary Fig. 5). CTAC is reported to form a less densely packed molecular layer on AuNRs than CTAB[27,28], allowing surface-initiated reactions to occur more readily. In addition, the higher reduction potential of $PtCl_4^{2-}$ compared to $PtBr_4^{2-}$ (0.75 V and 0.70 V for $PtCl_4^{2-}/Pt$ and $PtBr_4^{2-}/Pt$, respectively)[29,30] may further facilitate the surface-initiated redox reaction between the platinum and iron precursors.

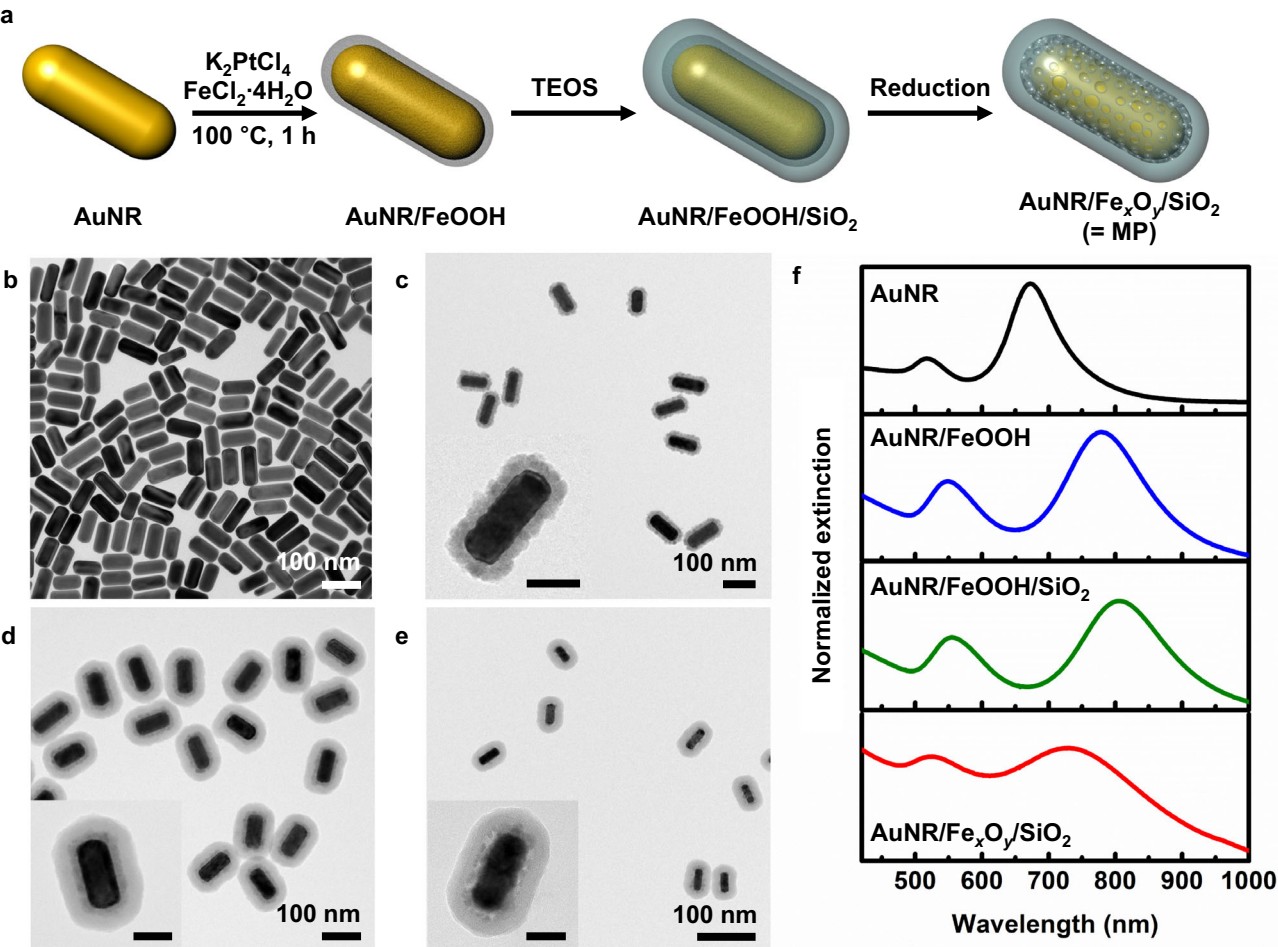

**Fig. 1 | Synthesis and characterization of MPs. a** Schematic of the MP synthesis procedure. TEM images of AuNR (**b**), AuNR/FeOOH (**c**), AuNR/FeOOH/SiO₂ (**d**), and AuNR/Fe$_x$O$_y$/SiO₂ (**e**) (scale bar in inset images: 50 nm). **f** Extinction spectra of AuNR (black), AuNR/FeOOH (blue), AuNR/FeOOH/SiO₂ (green), and AuNR/Fe$_x$O$_y$/SiO₂ (red). Source data are provided as a Source Data file.

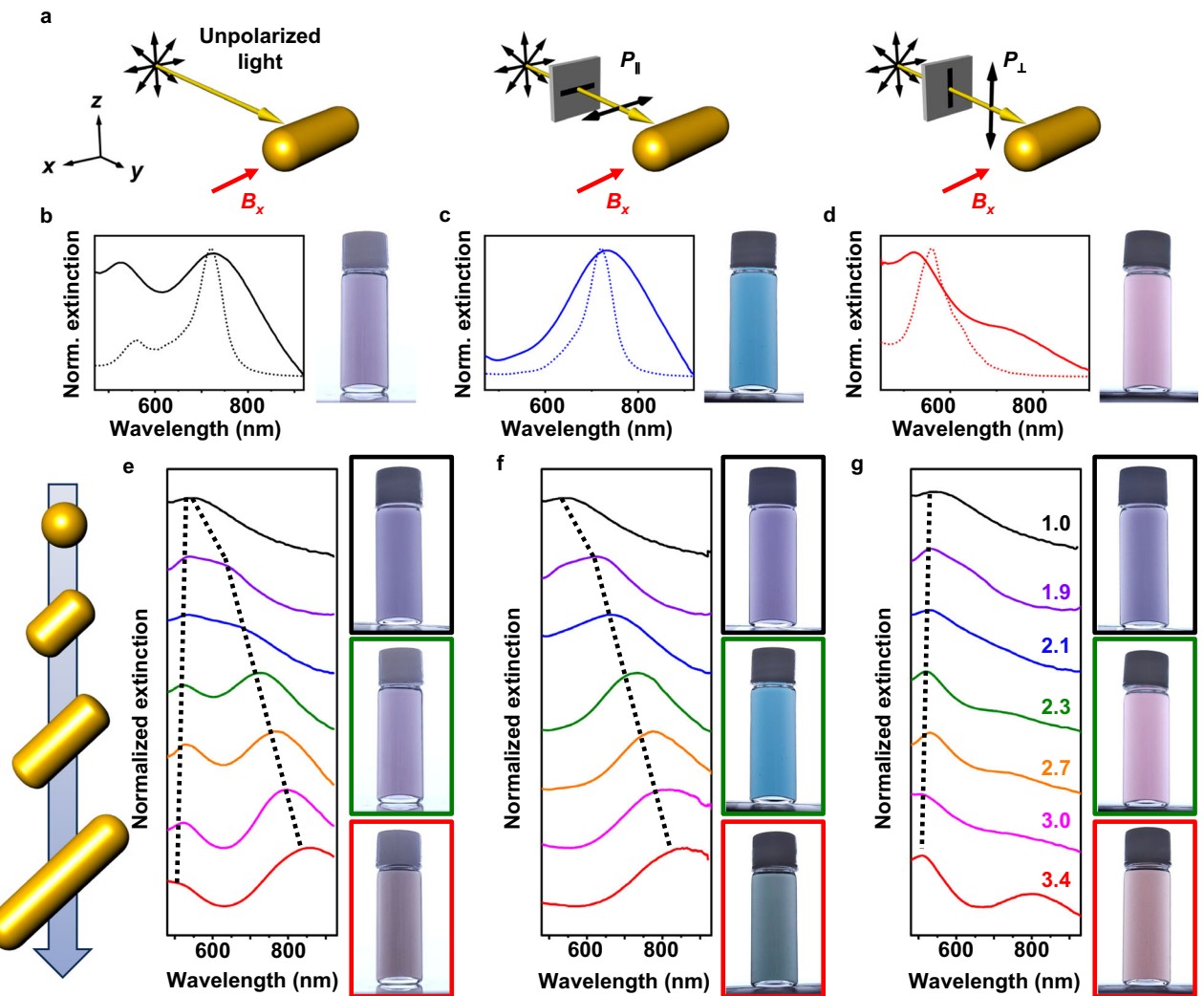

**Fig. 2 | Optical properties of MPs under $B_x$. a** Schematics depicting plasmonic excitation under unpolarized light, $P_\parallel$ and $P_\perp$. Black and red arrows indicate the direction of light polarization and applied magnetic field, respectively. Photographs and extinction spectra (measured: solid lines; simulated: dotted lines) of MPs incorporating AuNRs with an aspect ratio of 2.3 under unpolarized light (**b**), $P_\parallel$ (**c**), and $P_\perp$ (**d**). The experimental extinction spectra appear broader than the simulated spectra due to ensemble averaging over MPs with a size distribution. Extinction spectra and photographs of MPs containing AuNRs with varying aspect ratios under unpolarized light (**e**), $P_\parallel$ (**f**), and $P_\perp$ (**g**). The applied magnetic field strength was measured to be 18 mT. Source data are provided as a Source Data file.

The FeOOH-coated AuNR (AuNR/FeOOH) was further coated with $SiO_2$ (thickness: $21 \pm 2$ nm) using a modified Stöber method[31] to improve stability against high-temperature heat treatment during the following reduction (Fig. 1d)[32,33]. Thereafter, $SiO_2$-coated AuNR/FeOOH (AuNR/FeOOH/$SiO_2$) was thermally annealed under a high-temperature reductive condition (260 to 360 °C for 30 min to 2 h) to convert antiferromagnetic FeOOH into superparamagnetic $Fe_xO_y$. A porous $Fe_xO_y$ layer was observed between the AuNR core and the $SiO_2$ shell in the TEM image (Fig. 1e), with the porosity increasing with reduction time and temperature (Supplementary Fig. 6). X-ray diffraction (Supplementary Fig. 7) and X-ray photoelectron spectroscopy (Supplementary Fig. 8) analyses indicated the conversion of β-FeOOH to $Fe_xO_y$ comprising magnetite ($Fe_3O_4$) and maghemite (γ-$Fe_2O_3$). The superparamagnetic property of the resulting material was confirmed by magnetization measurements (Supplementary Fig. 9). The synthesized AuNR/$Fe_xO_y$/$SiO_2$ powder was dispersed in water by brief sonication (≈5 s). The aqueous dispersion was stable without aggregation, as confirmed by dynamic light scattering (DLS) analysis (Supplementary Fig. 10). Figure 1f presents the extinction spectra taken at each step of the synthesis. AuNRs with an aspect ratio of 2.3 exhibited T-LSPR and L-LSPR bands at 526 nm and 728 nm respectively. A

significant redshift in the L-LSPR band was observed upon FeOOH and $SiO_2$ coating, attributed to changes in the local refractive index (2.42 for FeOOH, 1.48 for $SiO_2$, and 1.33 for water)[34]. Upon reduction of FeOOH, a blue shift and broadening of the LSPR bands were observed, attributed to changes in refractive index due to the porous structure and to chemical interface damping by $Fe_xO_y$, respectively[35]. Reduction at high temperatures for extended durations enhanced the magnetic response but also caused significant LSPR damping. Therefore, the reduction conditions were optimized to achieve sufficient magnetic response without substantial damping of the LSPR band (Supplementary Fig. 11). The synthesized AuNR/$Fe_xO_y$/$SiO_2$, which is termed MP herein, oriented itself parallel to the direction of the magnetic field (Supplementary Fig. 12) to minimize the magnetic potential energy[36], enabling instant control over the alignment and arrangement of the plasmonic NPs.

## Optical properties of linearly aligned MPs

The optical properties of MPs can be controlled by selectively exciting T- or L-LSPRs through magnetic alignment (Fig. 2a). Figure 2b–d presents the extinction spectra of linearly aligned MPs under varying light polarization conditions. Under unpolarized light, both T- and L-LSPRs

are excited, producing two LSPR bands at 520 and 730 nm (Fig. 2b). Under linearly polarized light, T- or L-LSPRs can be selectively excited depending on the polarization direction of the light. Light polarization parallel to the applied magnetic field direction ($P_\parallel$) selectively excites the L-LSPR, resulting in an LSPR band at 730 nm (Fig. 2c). Light polarization perpendicular to the magnetic field direction ($P_\perp$) selectively excites the T-LSPR, resulting in an LSPR band at 520 nm (Fig. 2d). The solution color varied with the polarization direction of light, displaying blue and pink colors with $P_\parallel$ and $P_\perp$, respectively (Fig. 2c, d). The finite-difference time-domain simulation data of a model particle agreed well with the experimental results (Fig. 2b–d, dotted lines). As expected, the extinction spectra remain unchanged with the polarization state of light in the absence of a magnetic field (Supplementary Fig. 13). The rapid magnetic control of MP orientation was confirmed through spectral analysis under a rotating magnetic field (Supplementary Fig. 14 and Supplementary Movie 1).

A series of AuNRs with different aspect ratios were used to synthesize MPs to further verify the magnetic color tunability. Under unpolarized light, the extinction spectra showed a gradual redshift of L-LSPR bands as the aspect ratio increased from 1.9 to 3.4 (Fig. 2e). Selective excitation of the L-LSPR with $P_\parallel$ illumination displayed a blue-to-green color change as the aspect ratio increased (Fig. 2f), while the extinction maxima remained nearly constant with the selective excitation of T-LSPR through $P_\perp$ illumination (Fig. 2g). Gold nanospheres (AuNSs) without shape anisotropy displayed no spectral or color changes when a magnetic field was applied under varying polarization states of light (Fig. 2e–g (black); Supplementary Fig. 15). The experimental results for the series of MPs with different aspect ratios were in good agreement with the simulated spectra obtained from single-particle models (Supplementary Fig. 16).

An important advantage of our synthetic approach is its applicability to plasmonic NPs of nearly any size and shape, as it utilizes presynthesized NPs prepared by well-established methods. The generality

of our synthetic approach was further demonstrated using gold nanobipyramids (AuNBPs) and nanotriangles (AuNTs) (Supplementary Figs. 17 and 18). The extinction spectra of magnetically aligned 1D AuNBP/Fe$_x$O$_y$/SiO$_2$ were similar to those of AuNR/Fe$_x$O$_y$/SiO$_2$, displaying selective excitation of L-LSPR and T-LSPR under $P_\parallel$ and $P_\perp$, respectively (Supplementary Fig. 19a). AuNT/Fe$_x$O$_y$/SiO$_2$ with 2D geometry undergoes plane-parallel alignment under an external magnetic field (Supplementary Fig. 20)[15]. Consequently, $P_\parallel$ selectively excites the in-plane resonance, whereas $P_\perp$ excites both in-plane and out-of-plane resonances[37], leading to polarization-dependent optical properties (Supplementary Fig. 19b). Both NPs exhibited fast responses to changing magnetic fields (Supplementary Fig. 21). It is worth noting that our synthetic approach can generate relatively small size MPs whose orientational and translational movements can be separated under a mild magnetic field. This characteristic allows for the manipulation of the MP orientation without macroscopic phase separation.

## Birefringent properties of linearly aligned MPs

The anisotropic nature of the magnetically aligned MP ensemble can cause unusual optical phenomena associated with the polarization-dependence. Light transmission from the linearly aligned MPs was examined using different sets of cross-polarization conditions, where the input ($\theta_1$) and output ($\theta_2$) polarizer angles are described in Fig. 3. No light transmission was observed under the cross-polarization setting of $\theta_1 = 0°/\theta_2 = 90°$ or $\theta_1 = 90°/\theta_2 = 180°$ (Fig. 3a, b), as expected, since there is no mechanism for the electric field to rotate into alignment with the output polarizer. Conversely, a bright orange color appeared at the cross-polarization setting of $\theta_1 = 45°/\theta_2 = 135°$ and $\theta_1 = 135°/\theta_2 = 45°$ (Fig. 3c, d). This phenomenon arises from the birefringence of the MP arrays[38,39]. The linearly polarized incident electric field can be considered as the combination of two orthogonal components parallel and perpendicular to the long axis of the MP. In the case of $\theta_1 = 0°$ or $90°$ (Fig. 3a), the polarized light consists of only a

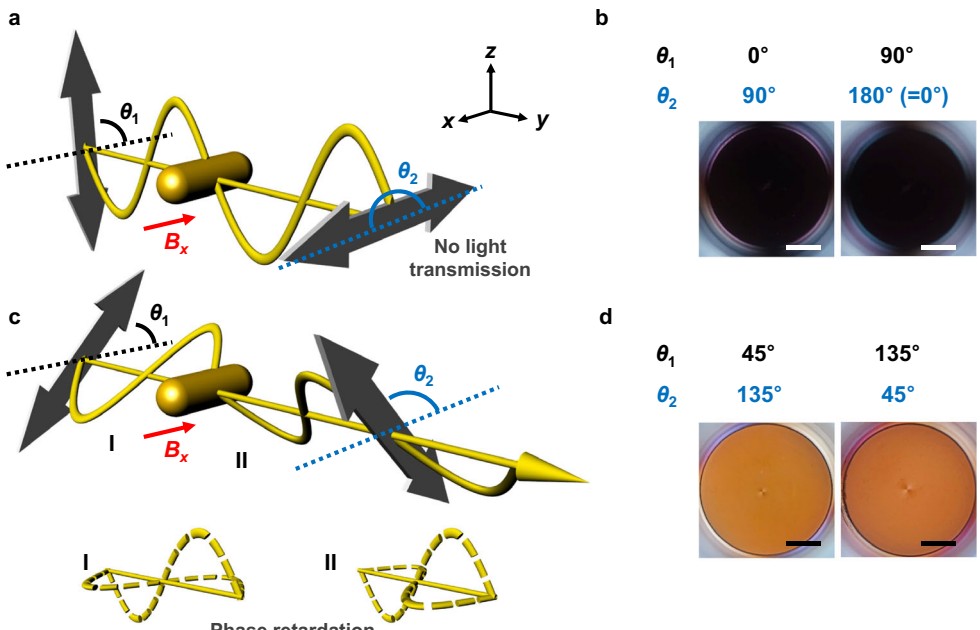

**Fig. 3 | Birefringent colors from linear MP arrays formed under $B_x$.** Schematic illustration (**a**) and photographs (**b**) showing no light transmission at the cross-polarization condition of $\theta_1 = 90°/\theta_2 = 180°$ or $\theta_1 = 0°/\theta_2 = 90°$, where $\theta_1$ and $\theta_2$ denote input and output polarizer angles relative to the magnetic field direction (red arrow, x-axis). Schematic illustration (**c**) and photographs (**d**) showing light transmission at the cross-polarization condition of $\theta_1 = 45°/\theta_2 = 135°$ or $\theta_1 = 135°/\theta_2 = 45°$. Flat gray arrows indicate the axes of polarizing filters. Yellow sinusoidal

lines represent the electric field waves of the incident light. A π phase shift between two orthogonal field components (dashed yellow lines in **c**) from region I to II results in a 90° rotation of the polarization direction. MPs containing AuNRs with an aspect ratio of 2.3 were used for the experiments. Photographs were taken under the applied magnetic field strength of 18 mT and white light illumination. Scale bars in all images represent 5 mm.

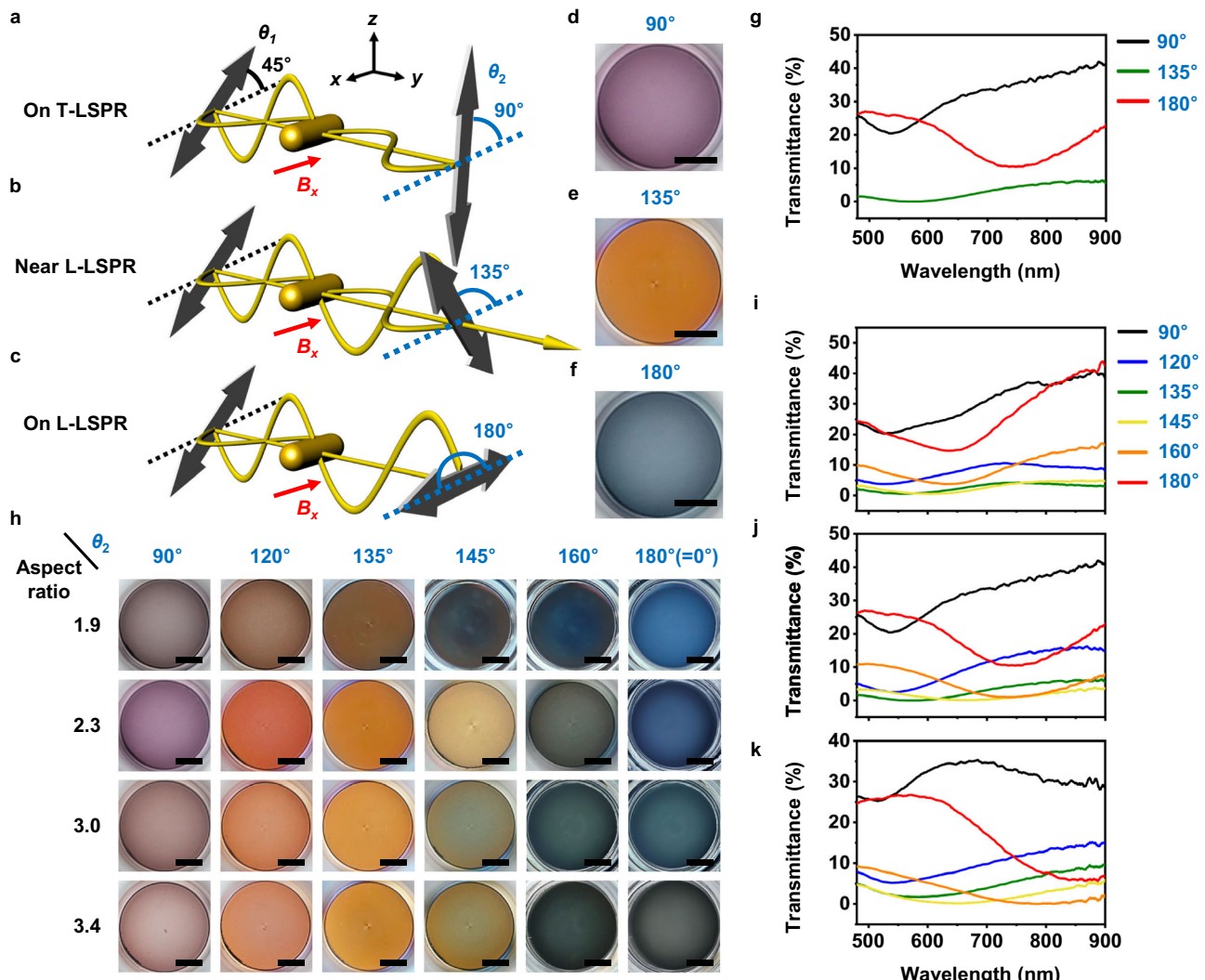

**Fig. 4 | Plasmon-mediated birefringent responses from linearly aligned MPs under varying $\theta_2$ at a fixed $\theta_1$ of 45°.** Schematics illustrating light transmission at $\theta_2 = 90°$ (**a**), 135° (**b**), and 180° (**c**) at T-LSPR, near L-LSPR, and L-LSPR wavelength, respectively. Red arrows indicate the direction of the applied magnetic field (x-axis). Flat black arrows indicate the axes of polarizing filters, where $\theta_1$ and $\theta_2$ denote polarizer angles relative to the magnetic field direction (dotted lines). Yellow sinusoidal lines represent the electric field wave of the incident light. Photographs of light transmission from linear MP (AuNR aspect ratio: 2.3) arrays at $\theta_2$ of 90° (**d**), 135° (**e**), or 180° (**f**). **g** Spectra of light transmission from linear MP (AuNR aspect ratio: 2.3) arrays at $\theta_2$ of 90° (black), 135° (green), or 180° (red). **h** Photographs of transmitted colors from linear MP arrays with varying AuNR aspect ratios under various $\theta_2$. Transmission spectra of linear MP arrays with AuNR aspect ratios of 1.9 (**i**), 2.3 (**j**), and 3.4 (**k**) collected at $\theta_2$ of 90°–180°. All experiments were conducted under the applied magnetic field strength of 18 mT and $\theta_1$ of 45°. Scale bars in all images represent 5 mm. Source data are provided as a Source Data file.

parallel or perpendicular component, which selectively excites the L- or T-LSPR respectively and is blocked by the cross polarizer at the output. For $\theta_1 = 45°$ or 135° (Fig. 3c), the MPs are excited simultaneously by the two orthogonal components of light (Fig. 3c, I). Each component excites its respective LSPR mode, accompanied by a phase retardation. If the phase difference between the field components is close to π, the polarization rotates by 90° (Fig. 3c, II), allowing light to transmit through the cross-polarizer.

To understand the birefringence further, we identify three key modes of transmission at $\theta_2 = 90, 135,$ and 180° when fixing $\theta_1 = 45°$, each resulting in a distinct color (Fig. 4a–f). As illustrated in Fig. 4a, the perpendicular component of light is selectively absorbed by the T-LSPR (540 nm), significantly reducing its intensity. Consequently, the field mostly consists of the parallel component, which is blocked when the output polarizer is aligned along the perpendicular direction (i.e., $\theta_2 = 90°$). A similar phenomenon occurs at the L-LSPR (750 nm, Fig. 4c), in which case light transmission is substantially reduced when the output polarizer is along the parallel direction (i.e., $\theta_2 = 180°$).

Accordingly, the transmission spectra presented in Fig. 4g demonstrate a pronounced valley at the T-LSPR and L-LSPR wavelengths for $\theta_2 = 90°$ and 180°, respectively. To understand the orange color observed under the cross-polarizer condition (Fig. 4b, e), we must consider the relative phases of the two orthogonal components. Since the incident electric field is composed of two orthogonal components, a phase shift in either component leads to a progression of polarization states—from linear to elliptical, to circular, and back to linear—with the final linear polarization exhibiting a different orientation (Supplementary Fig. 22). The phase difference is known to transition from 0 to π across the plasmonic resonance wavelength, amounting to π/2 at the resonance peak[38]. As the phase difference approaches π, the resultant electric field reorients from 45° to 135° (Supplementary Fig. 22), enabling strong transmission intensity through the cross-polarizer. Typically, the broader the resonance, the greater the offset between the wavelength at which the phase difference reaches π and the resonance wavelength[38]. In our MPs, this phenomenon is pronounced with the L-LSPR; the cross-polarized intensity (i.e., $\theta_2 = 135°$) appears at

a longer wavelength (≈850 nm) than the L-LSPR position (≈750 nm), as shown in Fig. 4g. This phase behavior leads to the appearance of colors distinct from those originating from the T- or L-LSPR absorption.

The plasmon-mediated birefringence was further investigated for MPs with varying aspect ratios. Figure 4h presents widely varying transmitted colors obtained by rotating $\theta_2$ from 90° to 180° at $\theta_1$ of 45°. Furthermore, different sets of color tones were obtained by changing the aspect ratio of MPs. As described above, colors with high T- or L-LSPR absorption contributions were observed at $\theta_2$ of 90° or 180°, respectively, and distinct birefringent colors appeared at $\theta_2$ near the cross-polarization condition. The transmission spectra gradually changed with $\theta_2$ (Fig. 4i–k; Supplementary Fig. 23), indicating that various colors can be realized through the plasmon-mediated birefringence (Supplementary Fig. 25). The dip position under the polarization setting of $\theta_2 = 180°$ shifted to longer wavelengths as the aspect ratio increased, confirming that the L-LSPR absorption effect is dominant under the polarization setting. These results demonstrate that plasmonic NPs with varying LSPR properties can be combined with the set-up capable of displaying birefringent effects such as our cross-polarized scheme to further expand the tunability of their optical properties.

As control experiments, additional transmission spectra were acquired with $\theta_1$ set at 0°, 90°, or 135° with varying $\theta_2$. The spectra were dominated by the L-LSPR or T-LSPR valleys when the input polarizer was set at $\theta_1 = 0°$ or $\theta_1 = 90°$, respectively (Supplementary Figs. 26 and 27). The data collected at $\theta_1$ of 135°, another condition of two polarization components sharing equal contributions, was similar to those at $\theta_1$ of 45° (Supplementary Fig. 28). When there was no input polarizer, only an L-LSPR or T-LSPR dip was observed at $\theta_2$ of 0° or 90°, respectively (Supplementary Fig. 29). In a randomly dispersed state without a magnetic field, birefringent colors did not appear; instead, the overall transmittance changed with $\theta_2$ (Supplementary Fig. 30). These results confirm that the polarization rotation by the aligned MPs is responsible for the light transmission and spectral changes under the cross-polarization setting of $\theta_1 = 45°$ or 135°.

### Extrinsic chirality of helically arranged MPs

Although individual MPs are achiral, their orientation and arrangement can be manipulated to induce circular birefringence, a phenomenon referred to as extrinsic chirality. As shown in Fig. 5a, b, a helical magnetic field was created by placing the sample between two bar magnets arranged at 45° or 135° from the light propagation direction ($y$-axis)[40]. This cross configuration of bar magnets generates a helical distribution of magnetic field across the sample, confirmed by the magnetic field simulation (Fig. 5c–e) where the magnetic field vectors show gradual rotation along the light propagation direction. The magnets at 45° creates a left-handed helical field in the sample position (Fig. 5d and Supplementary Movie 2), inducing left-handed helical assembly of MPs along the light propagation direction. Similarly, the magnets at 135° induces the right-handed helical assembly of MPs (Fig. 5e and Supplementary Movie 3). The helical arrangement of MPs was supported by dark-field optical microscopy (Supplementary Fig. 31). The left- and right-handed helical assemblies displayed strong positive and negative circular dichroism (CD) signals, respectively (Fig. 5f). As the aspect ratio of the AuNRs increased from 2.3 to 3.4, the CD peak position shifted from 580 to 660 nm, consistent with the trend of the L-LSPR wavelength. The helical superstructure possessed high optical asymmetry, exhibiting a CD of up to 12° and a $g$-factor of up to 0.21. The $g$-factor observed in this work is among the highest reported values for plasmonic systems[41–44] and constitutes the highest value reported thus far for dynamic assemblies of achiral NPs[13,14,16,17,25,40,45–54]. The CD signal was highly reproducible while switching between the left- and right-handed helical assemblies (Supplementary Fig. 32). Isotropic AuNS/Fe$_x$O$_y$/SiO$_2$ also displayed CD signals under a helical magnetic field, presumably due to the imperfection in the spherical shape and the formation of linear assemblies of NPs. However, its $g$-

factor was two orders of magnitude lower than that of AuNR/Fe$_x$O$_y$/SiO$_2$ (Supplementary Fig. 33). AuNR/Fe$_x$O$_y$/SiO$_2$ with a low aspect ratio (1.9) also exhibited noticeably lower chiroptical properties compared to those with higher aspect ratios (Supplementary Fig. 34). These results demonstrate that shape anisotropy significantly improves the chiroptical properties of magnetically aligned MPs.

## Discussion

In summary, we have developed an approach for synthesizing MPs composed of a plasmonic gold core and a superparamagnetic Fe$_x$O$_y$ shell. This method enables rapid, reversible, and highly flexible magnetic control of their optical properties. Under a linear magnetic field, the MPs align parallel to the direction of the magnetic field, enabling the selective excitation of the desired plasmonic mode depending on the polarization state of light. The linearly aligned MP arrays exhibit birefringence, displaying a wide variety of colors when viewed under input and output polarizers with varying angles. Furthermore, MPs aligned under a helical magnetic field exhibit strong chirality, with a $g$-factor of 0.21. The helical AuNR arrays showed much higher optical activities than AuNS arrays, indicating that the shape of the particle influences the chiral behavior. Importantly, the synthetic method is applicable to a wide range of NPs with different sizes and shapes, demonstrating its broad utility for plasmonic systems.

## Methods

### Materials

Gold(III) chloride trihydrate (HAuCl$_4$·3H$_2$O; ≥99.9%), CTAB (≥98%), sodium borohydride (NaBH$_4$; 99%), trisodium citrate, silver nitrate (AgNO$_3$; 99.9999%), L-ascorbic acid (reagent grade), CTAC (≥98%), citric acid (ACS reagent, ≥99.5%), sodium iodide (NaI; ACS reagent, ≥99.5%), trisodium citrate dihydrate, potassium tetrachloroplatinate(II) (K$_2$PtCl$_4$; 99.99%), iron(II) chloride tetrahydrate (FeCl$_2$·4H$_2$O; 98%), ammonium hydroxide solution (NH$_4$OH; 28-30%), poly(ethylene glycol) diacrylate (PEGDA, average $M_n$ 575 g mol$^{-1}$), $N,N'$-methylenebisacrylamide (≥99.5%), and glycerol (ReagentPlus®, ≥99.0%) were purchased from Sigma-Aldrich (St. Louis, MO, USA). Sodium oleate (NaOL; >97.0%) and tetraethyl orthosilicate (TEOS; >98.0%) were purchased from Tokyo Chemical Industry Co., Ltd. (Tokyo, Japan), Hydrochloric acid (HCl; 37% in water) was purchased from Daejung (Siheung, South Korea). Ethyl alcohol (99.9%) was purchased from Duksan (Seoul, South Korea). Ammonium persulfate was purchased from Bio-Rad (Hercules, CA, USA). All chemicals were used without any further purification. Water (18.2 MΩ cm) purified with Millipore Milli-Q system (Merck Millipore, Burlington, MA, USA) was used for all experiments.

### Characterization

TEM images were collected using JEOL JEM-2100 Plus with an accelerating voltage of 200 kV. Energy-dispersive X-ray spectroscopy (EDS) images were collected using JEM-ARM200F NEOARM. Extinction and transmission spectra were acquired with an Agilent 8453 spectrophotometer and an Ocean Optics QE Pro spectrometer. All photography images were taken with a front white LED lighting under white balance 6500 K and ISO 125 conditions. The camera exposure times were set at 8 ms, 20 ms, and automatic exposure conditions for unpolarized light, one polarizer, and two polarizers, respectively. Commercially available polarizing films (Sang-A Science, South Korea) were introduced at the front or back side of the sample as needed to create polarization conditions. X-ray diffraction (XRD) spectra were acquired by a Rigaku RINT 2100 diffractometer with Cu-$K_\alpha$ radiation ($\lambda = 0.154$ nm) at 30 mA and 40 kV. The surface chemical composition was analyzed using X-ray photoelectron spectroscopy (XPS; K-alpha, Thermo VG, U.K.) The magnetization curve was obtained with MPMS-XL5 at 300 K from −5 to 5 T. CD spectra were acquired using a JASCO J-1500 CD spectrometer. Magnetic field strength was measured using a

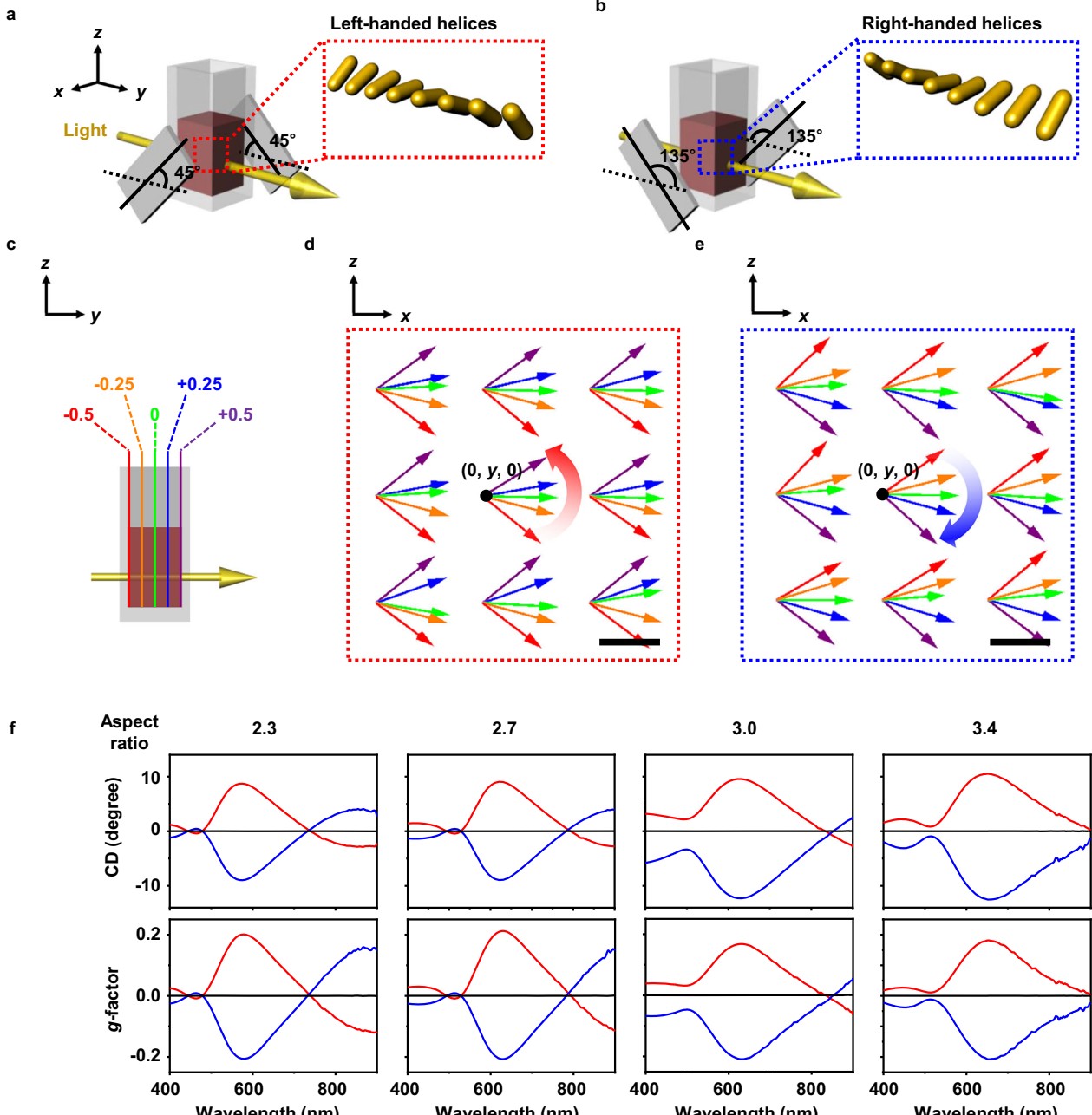

**Fig. 5 | Extrinsic chirality of helically arranged MPs.** Experimental set-ups used to induce left-handed (**a**) or right-handed (**b**) helical magnetic fields. Two bar magnets (gray flat rectangular bar) were positioned orthogonally in a cross configuration, with their axes oriented at 45° (**a**) and 135° (**b**) relative to the light propagation direction (yellow arrow, *y* axis). **c** Schematic illustration of the cross-sectional planes used in the magnetic field simulations. Each color represents a specific *y*-position along the light propagation direction (red: −0.5 cm; orange: −0.25 cm; green: 0 cm; blue: +0.25 cm; purple: +0.5 cm). The origin (0, 0, 0) is defined as the midpoint of the line connecting the centers of the two bar magnets. Simulated magnetic field vector plots corresponding to 45° (**d**) and 135° (**e**) cross configurations of bar magnets (NdFeB magnet, $20 \times 10 \times 1$ mm³) placed with an end-to-end separation distance of 1.5 cm. The arrows represent the magnetic field direction marked every 0.25 cm along the light propagation direction (*y* direction). The color of each vector corresponds to a specific *y*-position marked in (**c**). The black dot represents the central point of the simulation plane at (*x*, *z*) = (0, 0). The red and blue curved arrows indicate the rotational orientation of the magnetic field with respect to the light propagation direction. The variations in the apparent lengths of the magnetic field vectors arise from their tilt relative to the *x*–*z* plane. The scale bars represent 200 μm for both plots. **f** CD and *g*-factor spectra of MPs with AuNR aspect ratios of 2.3, 2.7, 3.0, and 3.4 under helical magnetic fields (20 mT) with 45° (red) and 135° (blue) magnet settings along with the data collected without a magnetic field (black). Source data are provided as a Source Data file.

455 DSP Gauss meter. Dark-field optical microscope images were captured using an Olympus BX53M microscope equipped with a halogen lamp, a dark-field condenser, and a 50× MPlanFL N objective lens (NA = 0.80). Dynamic light scattering (DLS) measurements were performed using a Malvern Panlytical Zetasizer Nano-ZS analyzer equipped with a 632.8 nm laser at a scattering angle of 173°.

## Synthesis of AuNR with aspect ratios of 1.9 and 2.1

AuNRs with aspect ratios of 1.9 and 2.1 were prepared using a seed-mediated growth method[55]. To prepare the seed solution, 5 mL of 0.5 mM HAuCl₄·3H₂O was mixed with 5 mL of 0.2 M CTAB solution in a 20 mL vial. Next, 0.6 mL of fresh 0.01 M NaBH₄ diluted with 0.4 mL water was added into the HAuCl₄/CTAB solution under rapid stirring at

1200 rpm. After 2 min stirring, the seed solution was aged at room temperature for 3 h before use. To prepare the growth solution, 0.9 g of CTAB and 0.08 g of sodium salicylate were dissolved in 25 mL of warm water (≈50 °C) and then cooled to 30 °C. Subsequently, 600 μL of 4 mM $AgNO_3$ solution was added, and the mixture was kept at 30 °C for 15 min. Thereafter, 25 mL of 1 mM $HAuCl_4·3H_2O$ was added while stirring slowly (400 rpm). After 15 min, 100 μL of 64 mM L-ascorbic acid was added, and the solution was stirred vigorously for 30 s until it became colorless. For AuNRs with aspect ratios of 1.9 and 2.1, 50 μL and 80 μL of the seed solution were added into the growth solution, respectively. The resulting mixture was stirred for 30 s and left at 30 °C for 12 h. The reaction product was collected via centrifugation at 8030 × $g$ for 10 min. AuNR concentration was estimated from the extinction spectra using an extinction coefficient[56] of 9.1 × $10^8$ M $cm^{-1}$ at the longitudinal peak position for AuNRs with the aspect ratio <3.

### Synthesis of AuNRs with aspect ratios of 2.3, 2.7, 3.0, and 3.4

AuNRs with various aspect ratios were prepared using a seed-mediated growth method[57]. The seed solution was prepared using the method described in the previous paragraph. To prepare the growth solution, 0.7 g of CTAB and 0.1234 g of NaOL were dissolved in 25 mL of warm water (≈50 °C) and then cooled to 30 °C. Subsequently, 1.8 mL of 4 mM $AgNO_3$ solution was added, and the mixture was kept at 30 °C for 15 min. Thereafter, 25 mL of a 1 mM $HAuCl_4 · 3H_2O$ solution was added and stirred at 700 rpm for 90 min. The pH was then adjusted by introducing 150 μL of HCl (37 wt.%, 12.1 M in water). After stirring gently at 400 rpm for 15 min, 125 μL of 0.064 M L-ascorbic acid was added, and the solution was stirred at 1200 rpm for 30 s. Depending on the target aspect ratio, various amounts of seed solution ranging from 100 μL to 2 mL were injected into the growth solution with rapid agitation. The resulting mixture was stirred for 30 s and left at 30 °C for 12 h. The final product was isolated by centriguation at 5441 × $g$ for 30 min and redispersed in water. AuNR concentration was estimated from the extinction spectra using an extinction coefficient of 1.1 × $10^9$ M $cm^{-1}$ at the longitudinal peak position for AuNRs with the aspect ratio of 3–4[56].

### Synthesis of AuNS

AuNSs were synthesized using a seed-mediated growth method[58]. To obtain seed particles, 0.6 mL of fresh 10 mM $NaBH_4$ solution was rapidly added to a 10 mL aqueous solution containing $HAuCl_4·3H_2O$ (0.25 mM) and CTAB (100 mM). The mixture was placed on an orbital shaker at a speed of 300 rpm for 2 min and then kept undisturbed at 27 °C for 3 h. To prepare the growth solution for 12 nm AuNSs, 4 mL of 0.5 mM $HAuCl_4·3H_2O$ and 3 mL of 100 mM L-ascorbic acid aqueous solution were sequentially added to 4 mL of a 200 mM CTAC solution. After 5 min, 0.1 mL of gold seeds was added into the growth solution under vigorous stirring. After stirring for 30 s, the solution was left on the shaker at room temperature for 30 min. The particles were collected via centrifugation at 18,748 × $g$ for 30 min and then redispersed in 2 mL water. AuNSs with an average diameter of 40 nm were synthesized using the 12 nm AuNSs as seeds. Specifically, 0.52 mL of 10 mM L-ascorbic acid and 64 μL of the 12 nm AuNSs were mixed in 8 mL of 100 mM CTAC, and then 2 mL of 2 mM aqueous $HAuCl_4·3H_2O$ was added five times at a rate of 0.333 mL every 10 min. The reaction was allowed to proceed at room temperature for 2 h. The product was collected through centrifugation at 11,103 × $g$ for 10 min, washed once with water, and redispersed in water for further use.

### Synthesis of AuNBP

AuNBPs were synthesized using a seed-mediated growth method[59,60]. The seed solution was prepared by adding 0.25 mL of 0.01 M $HAuCl_4·3H_2O$ and 0.25 mL of 0.2 M citric acid solutions into 10 mL of a 0.05 M CTAC solution under magnetic stirring (700 rpm). Subsequently, 0.125 mL of fresh 0.05 M $NaBH_4$ was added to the mixture

while vigorously stirring it at room temperature. After stirring for 2 min, the resultant solution was aged at 100 °C for 90 min. After aging, the color of the seed solution changed from brown to dark red. The quality of the seed solution has a significant impact on the purity of the synthesized particles. Therefore, when making a seed solution, it is advisable to prepare all solutions, including water, in a fresh state immediately before synthesis. The growth solution was prepared by adding 2 mL of 0.01 M $HAuCl_4·3H_2O$, 0.4 mL of 0.01 M $AgNO_3$, and 0.8 mL of 1 N HCl solution into 40 mL of 0.1 M CTAB solution with stirring at 700 rpm and then adding 0.32 mL of 0.1 M L-ascorbic acid solution to the mixture at the stirring speed of 1200 rpm. When the mixture color changed from yellow to colorless, 2 mL of the seed solution was added to the mixture. After 10 s of stirring, the solution was left undisturbed for 4 h in a 30 °C bath. The AuNBPs were washed via centrifugation (8029 × $g$, 10 min) and redispersed in 20 mL of water for further use.

### Synthesis of AuNT

AuNTs were synthesized using a seed-mediated overgrowth synthesis method reported in ref. 61. To prepare the seed solution, 25 μL of 50 mM $HAuCl_4·3H_2O$ was added to 4.7 mL of a 100 mM CTAC solution, and the mixture was stirred at 300 rpm for 5 min. Next, 300 μL of fresh 10 mM $NaBH_4$ solution was rapidly added into the vial under vigorous stirring, and the solution was left undisturbed for 2 h under mild stirring at room temperature. Solution A used for the overgrowth of gold mono-twinned seeds was prepared by sequentially adding 1.6 mL of 100 mM CTAC, 40 μL of 50 mM $HAuCl·3H_2O$ solution, and 15 μL of 10 mM NaI solution into 8 mL of water. Solution B used for the final growth step to yield AuNTs was prepared by sequentially adding 20 mL of 100 mM CTAC solution, 500 μL of 50 mM $HAuCl·3H_2O$, and 300 μL of 10 mM NaI into 20 mL of water. Subsequently, 40 μL of 100 mM L-ascorbic acid was added to solution A, and 400 μL of 100 mM L-ascorbic acid was added to solution B under vigorous stirring. Thereafter, 100 μL of the 10-fold diluted seed solution was added to solution A, and 3.2 mL of solution A was quickly transferred to solution B. Next, solution B was left undisturbed at room temperature for 1 h. Afterward, 630 μL of 25 wt.% CTAC solution was added to the solution, which was then left overnight to allow the nanotriangles to settle at the bottom of the flask. The pink supernatant was carefully removed, and the precipitate was redispersed in 5 mL of 20 mM CTAC solution.

### Synthesis of AuNR/FeOOH

CTAB-coated AuNRs (4.4–6.6 nM) were washed with 5 mL of 0.15 mM CTAC solution to replace CTAB with CTAC. Subsequently, the AuNR solution was mixed with 300 μL of a 0.4 mM $K_2PtCl_4$ solution and left for 2 min to allow $PtCl_4^{2-}$ to adsorb onto the AuNRs. Thereafter, 500 μL of a freshly prepared $FeCl_2·4H_2O$ (10–400 mM) solution and 4.2 mL of water were sequentially added to the AuNR solution with gentle shaking. The solution was kept at 100 °C for 1 h to allow the iron precursor to oxidize into FeOOH. The final products were collected via centrifugation at 7120 × $g$ for 10 min and redispersed in 1 mL water for further use.

### Synthesis of AuNR/FeOOH/SiO$_2$

To protect AuNR/FeOOH from high-temperature conditions during the subsequent reduction process[32,33], AuNR/FeOOH was coated with a $SiO_2$ shell using a modified Stöber method[31]. Specifically, 5 mL of the AuNR/FeOOH solution was mixed with 20 mL of ethanol under gentle stirring. Subsequently, 300 μL aqueous solution of ammonium hydroxide was added into the solution to adjust pH. Finally, 250 μL of TEOS was injected into the solution under vigorous stirring. After 30 min, AuNR/FeOOH/$SiO_2$ was collected via centrifugation at 7120 × $g$ for 5 min and redispersed in 20 mL of ethanol. The washing process was repeated two times, and the product was dried at 60 °C overnight.

### Thermal reduction of FeOOH to $Fe_xO_y$

AuNR/FeOOH/SiO$_2$ powder was heated at 310 °C for 2 h in a hydrogen atmosphere (Ar/H$_2$, 95%/5%) in a tube furnace to reduce FeOOH into $Fe_xO_y$[62]. After the reaction, the tube furnace was allowed to cool to room temperature under continuous gas flow. The resulting AuNR/$Fe_xO_y$/SiO$_2$ powder was collected after ≈5 h and dispersed in water by brief sonication (≈5 s).

### Synthesis of AuNS/$Fe_xO_y$/SiO$_2$, AuNBP/$Fe_xO_y$/SiO$_2$, and AuNT/$Fe_xO_y$/SiO$_2$

AuNS/$Fe_xO_y$/SiO$_2$, AuNBP/$Fe_xO_y$/SiO$_2$, and AuNT/$Fe_xO_y$/SiO$_2$ were synthesized using the same procedure as AuNR/$Fe_xO_y$/SiO$_2$, but the NP, K$_2$PtCl$_4$, and FeCl$_2$·4H$_2$O concentrations were slightly adjusted to obtain a uniform shell. For FeOOH coating, the NP concentration was adjusted based on the main plasmon peak intensity in the extinction spectrum, which was adjusted to 1.0, 3.0, and 3.0 nM for the AuNSs, AuNBPs, and AuNTs, respectively. For the AuNTs, the K$_2$PtCl$_4$ and FeCl$_2$·4H$_2$O concentrations were adjusted to 0.8 mM and 20 mM, respectively. Regarding the AuNBPs, SiO$_2$ coating was performed using high NP concentrations of 6.0 nM. All processes not mentioned followed the same protocol as the AuNR/$Fe_xO_y$/SiO$_2$ synthesis method.

### Finite-difference time-domain (FDTD) modeling

Numerical calculations were performed using commercial FDTD software (Lumerical FDTD, Release 2018a, Version 8.19.1584; Lumerical Inc., Canada) to calculate the extinction cross-section of AuNS/$Fe_xO_y$/SiO$_2$ and AuNR/$Fe_xO_y$/SiO$_2$ nanostructures. The simulations were performed on a AuNR/$Fe_xO_y$/SiO$_2$ and six representative AuNR/$Fe_xO_y$/SiO$_2$ with different NP geometries, as summarized in Supplementary Table 1. Each sample consisted of a AuNS or AuNR with specific dimensions and aspect ratio, coated with $Fe_xO_y$ and SiO$_2$ shells of varying thicknesses. The modeled structures reflect experimentally synthesized NPs whose dimensions were measured by TEM analysis. In some cases, the NP dimensions were slightly adjusted within the TEM-estimated error range to better match the experimentally observed extinction peaks. The optical constants were obtained from previously reported data, using values from Johnson and Christy[63] for gold, from Palik[64] for SiO$_2$, and a constant refractive index of 2.3 for $Fe_xO_y$ across the relevant wavelength range[65]. The surrounding medium was set to water with a refractive index of 1.33. A total-field scattered-field (TFSF) source was applied over the 480 to 900 nm wavelength range. To capture the polarization-dependent plasmonic response, two excitation conditions were used, one with the electric field aligned parallel to the nanorod's long axis to excite the $P_\parallel$, and the other with the field perpendicular to it to excite the $P_\perp$. A conformal mesh with a minimum resolution of 0.5 nm was applied near the nanostructures to ensure sufficient spatial accuracy. The simulation domain was enclosed by perfectly matched layers in all directions to eliminate artificial reflections at the boundaries. The extinction cross-sections were obtained by integrating the absorbed and scattered power throughout the simulation region.

### Optical measurements under two polarizers

The measurements were carried out in a black box with openings on the top and bottom faces covered with polarizing films placed on a white LED light pad (Comzler A4 LED Light Board) used as the light source (Supplementary Fig. 24). A linear magnetic field was applied using a nickel-plated neodymium (NdFeB) magnet (50 × 20 × 10 mm$^3$). The input polarization angle ($\theta_1$) was 45° with respect to the magnetic field direction, and images were obtained while changing the angle of the output polarization ($\theta_2$).

### Simulation of the magnetic field

Ansys Maxwell Electromechanical Device Analysis Software (Release 2024 R2) was used to simulate the magnetic field distribution for two different magnet settings used to induce left-handed and right-handed helical magnetic fields, in which two bar magnets (NdFeB magnet, 20 × 10 × 1 mm$^3$) were positioned orthogonally in a cross configuration, with their axes oriented at 45° (left-handed helical field) or 135° (right-handed helical field) relative to the light propagation direction, and placed with an end-to-end separation of 1.5 cm (Fig. 5a, b). NdFeB-N35 was used as the magnet material, and air was chosen as the ambient condition. To visualize the spatial distribution of the magnetic field, field distribution was extracted on cross-sectional planes perpendicular to the light propagation direction (optical axis) at 0.25 cm intervals (Fig. 5c). These slices were subsequently overlaid to reconstruct a three-dimensional field representation, as shown in Fig. 5d, e.

### Hydrogel fixation of AuNR/$Fe_xO_y$/SiO$_2$ alignment under a helical magnetic field

To preserve and visualize the helical arrangement of MPs, the particles were immobilized in a polyacrylamide-based hydrogel. Specifically, 0.2 g of acrylamide, 0.19 mL of acrylic acid, and 25 mg of N,N'-methylene bisacrylamide were dissolved in 0.125 mL of water with sonication. Subsequently, 0.875 mL of glycerol was added to the solution and the mixture was sonicated for 10 s. Then, 20 mg of ammonium persulfate, acting as an initiator for polymerization, was added to 1 mL of the monomer mixture. MPs were then dispersed in the solution, referred to as the pre-gel solution, at a concentration of 1 mg mL$^{-1}$. To cast the gel, the MP-containing pre-gel solution was injected between two glass slides with a ≈1 mm PDMS spacer. During the gelation, a helical magnetic field was applied by placing two bar magnets in the cross configuration described above. The gelation was initiated by exposing the pre-gel solution to UV light (365 nm) for 2 min, which fixed the magnetic alignment of MPs within the hydrogel.

## Data availability

The data that support the findings of this study are available from the corresponding authors upon request. Source data are provided with this paper.

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

## Acknowledgements

We acknowledge the financial support received from the National Research Foundation of Korea (NRF) (RS-2024-00397807 and RS-2023-00274736) grant funded by the Korean government (MSIT). This work was also supported by the Korea Basic Science Institute (National Research Facilities and Equipment Center) grant funded by the Ministry of Education (2020R 1A 6C 101B194).

## Author contributions

H.K., Y.J., and S.-J.P. conceived and designed the study. H.K. and Y.J. performed the primary investigation and formulated the research hypothesis. H.K., Y.J., K.B., S.P., J.L., T.S.S., J.K.H., and S.-J.P. conducted experimental investigation, data collection, and data interpretation. H.K., Y.J., and S.-J.P. wrote the original draft of the manuscript. T.S.S., J.K.H., and S.-J.P. reviewed and edited the manuscript. S.-J.P. supervised the overall study and provided guidance throughout the project. S.-J.P. secured the funding necessary for the study. All authors contributed to the writing and review of the manuscript.

## Competing interests

The authors declare no competing interests.
