## [Transparent Peer Review file · Nature Communications]

Dynamic birefringence and chirality of magnetically controllable assemblies of anisotropic plasmonic nanoparticles in dispersion

Corresponding Author: Professor So-Jung Park

Version 0:

Reviewer comments:

Reviewer #1

(Remarks to the Author)

This manuscript reports the synthesis and optical properties, including strong chiroptical response of magneto-plasmonic nanoparticles of a few different shapes of gold nanoparticles, mostly focusing on gold nanorods. This work is comprehensive and included extensive Supporting Information. Two significant advances in this manuscript merit publication in Nature Communications: 1. The approach for the magnetic overcoating assisted by a Pt salt is novel and useful. 2. These nanoparticles have a strong chiroptical response. A potential challenge with this work is that there appears to be some damage to the plasmonic nanoparticles during the processing steps. This is evident in Fig. 1f, where the ratio of the L-SPR/T-SPR intensities successively decreases, and both SPRs broaden. Fig. S5 shows that the plasmonic core is sensitive to higher annealing temperatures and times, which are necessary to obtain significant alignment in magnetic fields, shown in Fig. S9. Therefore, there is a tradeoff between narrow SPR bands and strong responses to magnetic fields. This work may still be suitable for Nature Communications, but this issue does somewhat reduce the impact.

1. I recommend adjusting the title. Tunable or Dynamic would be more accurate than Reconfigurable, since no configurations are locked into place.
2. The rationale for use of K₂PtCl₄ and why exchange of CTAB with CTAC is important should be discussed in more detail.
3. After performing annealing to obtain the AuNR/IO/SiO₂ nanoparticles, they were redispersed through sonication in water. Was any sintering between nanoparticles observed, and did they redisperse well?
4. The experimental results through crossed polarizers are quite interesting, but the discussion of birefringence should be improved. How can the authors distinguish between birefringence and linear dichroism? This sentence mentions scattering: "If the phase difference between the two scattered components is close to..." Why is scattering emphasized over absorption? Given the small size of the nanorods, I would expect relatively little scattering, and absorption would be the main mechanism of extinction.
5. For comparison, what are the highest CD angle and g-factor previously reported for chiral assemblies of achiral plasmonic nanoparticles?
6. More references for magnetically functionalized gold nanorods and their alignment in magnetic fields should be added.

Reviewer #2

(Remarks to the Author)

The authors developed a new synthetic approach for magnetically controllable plasmonic nanoparticles (MPs) composed of an anisotropic gold core and an iron oxide (IO) shell, enabling rapid magnetic alignment control while preserving tunable plasmonic properties. Linearly aligned MPs exhibit reconfigurable birefringence with polarization-dependent color tuning, while helically arranged MPs achieve record-breaking extrinsic chirality (g-factor = 0.21) through magnetic field-induced assembly of superstructures. The result shows broad applicability for smart optical devices requiring real-time plasmonic

and chiral modulation. I would recommend the publication of this manuscript on Nature Communications after minor revisions.

1. Could the authors provide experimental data (e.g., XRD and XPS) obtained under different thermal treatment conditions to clarify the transformation from IOH to IO and confirm the phase composition of the resulting IO?
2. Could the author explain the origin of the chiral optical response observed in isotropic gold nanospheres under helical magnetic fields?
3. Could the authors explain why, as the aspect ratio of the nanorod structure increases from 2.3 to 3.4, the circular dichroism of the helically arranged MPs becomes larger, while the g-factor tends to decrease or remain unchanged (Fig. 5c)?
4. Are there any other methods to measure the orientation of MPs except by directly calculating the direction of the magnetic field? Does the measurement result of the chiral optical response depend on the illumination direction in the authors' experiments?

Reviewer #3

(Remarks to the Author)

The controlled production of nanoparticles with reconfigurable optical activity is certainly a topic of great interest for a broad audience, which makes this manuscript potentially adequate for the Nature Communications readership after some improvements are done.

While the synthetic approach and the post-synthetic treatment of the different plasmonic nanoparticles are described in detail, the protocols, both for experiments and simulations, are too succinct to allow the replication of the results and the lacking information also makes it difficult to understand the underlying physics.

For instance, the electrodynamic simulations contain no description at all of the size and shape of the model nanoparticles, the authors only say what software has been used and the dielectric constant of each material, which would not be enough if I wanted to replicate their FDTD simulations.

But what I missed the most was a more clear and detailed description of how the magnetic field was applied to the samples in each experiment, which is the core of this investigation. Beginning with the TEM characterization depicted in Fig. S10, the illustration has no description of the experimental setup, authors should indicate clearly what each graphical element stands for. Scheme S1 has a more complete description of the elements comprising the experimental setup, but it does not indicate where the magnets are. I might assume the magnet is the gray block on the left of the sample, but if this is the case then the magnetic field is not homogeneous across the samples, as would be the case for the pair of parallel planar magnets depicted in Fig. S10 (assuming that the red/blue cylinders are the magnets). The same issues apply to Fig. 5 and Fig. S27, which mention the alignment of the magnets with respect to the direction of light, but the cartoon misses proper description and the text is also unclear about what these angles mean. It is also unclear what are the magnetic dipoles depicted in Figs. 5b and S27 - are they the magnetic dipoles of each nanoparticle? or are they projections of the magnetic field between the magnets? In either case, this takes me back to the description of the simulations of the magnetic fields, which is also too succinct to allow replication, the SI text only says what software was used and the dimensions of the magnets.

At this point, I would not analyze the results in depth because of the many doubts I have about the experimental and computational setups, so I request that authors should deepen the description of each and every experiment and simulation. If I may suggest, authors should include in all cartoons vectors depicting the direction of the magnetic field, since this is the origin of the reconfigurable optical activity being discussed all along the manuscript. These details matter for the broader audience that awaits ahead, since this broader audience is much less familiar with the topics being discussed than the expert reviewers handling the manuscript at this stage.

Reviewer #4

(Remarks to the Author)

The manuscript authored by Kang delineates the synthesis and dynamic (chir)optical properties of magneto-plasmonic nanorods (MP) manipulated by magnetic field. The MPs synthesized by the growth of iron oxide on the surface of Au nanorods show excellent superparamagnetic properties, which can be aligned by an external magnetic field. Owing to the dual resonance modes inherent to the gold nanorods, the aligned MPs exhibit birefringence property under unidirectional linearly polarized light. More interestingly, MPs can be aligned by helical magnetic field to produce chiral assembly of MPs with outstanding chiroptical properties with a g-factor up to 0.21. I appreciate the high innovation of this work and the clarity of the data analysis. I would suggest this work for publication only after a minor revision.

1. The authors claimed in the manuscript that this work has achieved the highest g-factor (0.21) for a dynamical dynamic nanoparticle assemblies to date. However, to the best of my knowledge, a recent work on the chiral assembly of chiral gold nanorods has reported a g-factor of 0.24 (Nano Lett. 2024, 24, 41, 13027-13036). Therefore, the authors need to revise their statement, although it would not affect the impact of the work.
2. Please indicate the direction of the magnetic field in Fig. 3 and Fig. 4, as those shown in Fig. 2, and emphasize the distinction between the magnetic field and the optical field in the figures.
3. What was the strength of the applied magnetic fields?
4. I really would like to recommend the authors to use Fe_xO_y to replace IO, which is hardly be used in the field of chemistry or materials.

5. For the helical magnetic field alignment, I am wondering what were the extinction spectra of the MP solution? Were the nanorods aligned in a helical way as shown in the cartoon of Figure 5b? If so, what was the pitch distance and interrod distance?

Version 1:

Reviewer comments:

Reviewer #1

(Remarks to the Author)

The authors have comprehensively addressed the reviewers' comments and appropriately revised the manuscript. I recommend it for publication after addressing the following interrelated comments.

Can the MPs be trapped if a permanent magnet is placed next to a vial containing a dispersion of them, i.e., do they exhibit some level of magnetophoresis? It would be good to mention that for comparison with related magnetoplasmonic nanoparticles. That might also affect the extent to which a magnetic field gradient in Scheme S2 is negligible or not. Does the vector field from the simulations in Scheme S2 indicate the magnitude of the magnetic field in addition to the direction? It could be helpful to calculate the strength of the magnetic field toward the left-center and right-center of those simulations to confirm that the gradient is weak. Why is the simulated field asymmetrical? On the right side, some of the arrows tilt up (characteristic with the field of a magnetic dipole), but none tilt down (which would also be expected if the simulations are performed along a central axis).

Reviewer #2

(Remarks to the Author)

The manuscript has been improved a lot after the revision. The authors have addressed the critical aspects in the original manuscript. I believe that the manuscript can now be accepted in Nat. Comm..

Reviewer #3

(Remarks to the Author)

The authors have properly addressed all the questions I had risen in my first report and now the revised manuscript makes a clear description of all experimental setups and numerical simulations, allowing the results to be rationalized and reproduced. The findings are very interesting indeed, with optical properties being controlled by changes in size and shape during the synthesis and also being tuned post-synthetically by external magnetic fields. The results show that different combinations of the magnetic field and the polarization of light give rise to different optical properties, including the strong circular dichroism under helical magnetic fields. All in all, I consider this manuscript should be published as it is in Nature Communications.

Reviewer #4

(Remarks to the Author)

The authors have answered all my questions and made the corresponding revision to strengthen the manuscript. I have no more comments but hope this manuscript to be published as soon as possible.

Response to reviewers' comments

We thank the reviewers for their positive and helpful comments. We have revised the manuscript accordingly. In brief, we conducted additional experiments, provided new data, and clarified the text to address the reviewers' concerns. These revisions have significantly strengthened the manuscript. Detailed responses to each comment are provided below. Please also refer to the marked copies of the manuscript and supporting information, in which all changes are highlighted in red.

Reviewer 1

This manuscript reports the synthesis and optical properties, including strong chiroptical response of magneto-plasmonic nanoparticles of a few different shapes of gold nanoparticles, mostly focusing on gold nanorods. This work is comprehensive and included extensive Supporting Information. Two significant advances in this manuscript merit publication in Nature Communications: 1. The approach for the magnetic overcoating assisted by a Pt salt is novel and useful. 2. These nanoparticles have a strong chiroptical response. A potential challenge with this work is that there appears to be some damage to the plasmonic nanoparticles during the processing steps. This is evident in Fig. 1f, where the ratio of the L-SPR/T-SPR intensities successively decreases, and both SPRs broaden. Fig. S5 shows that the plasmonic core is sensitive to higher annealing temperatures and times, which are necessary to obtain significant alignment in magnetic fields, shown in Fig. S9. Therefore, there is a tradeoff between narrow SPR bands and strong responses to magnetic fields. This work may still be suitable for Nature Communications, but this issue does somewhat reduce the impact.

Response: We sincerely thank the reviewer for the positive and constructive feedback. As pointed out, there exists a trade-off between SPR bandwidth and magnetic responsivity. Nevertheless, magneto-plasmonic nanoparticles with balanced optical and magnetic properties can be synthesized under appropriate conditions, as mentioned in our originally submitted manuscript. In future work, we aim to further address this issue by introducing a thin silica spacer between the plasmonic core and the magnetic shell.

Comment 1: 1. I recommend adjusting the title. Tunable or Dynamic would be more accurate than Reconfigurable, since no configurations are locked into place.

Response: Following the reviewer's suggestion, we changed "Reconfigurable" to "Dynamic" in the title.

Comment 2: The rationale for use of K_2PtCl_4 and why exchange of CTAB with CTAC is important should be discussed in more detail.

Response: We thank the reviewer for the helpful comment. As shown in Figure S2 of our originally submitted manuscript, a small amount of K_2PtCl_4 was essential for the formation of a uniform iron oxyhydroxide (FeOOH) shell on the surface of Au nanorods (AuNRs). The redox reaction between PtCl_4^{2-} and Fe^{2+} , facilitated by the gold surface, initiates the formation of an FeOOH layer on the AuNR surface. Further growth of the FeOOH shell occurs via the oxidation of Fe^{2+} by dissolved oxygen, resulting in AuNR/ FeOOH core/shell particles.

Regarding the surfactant exchange, CTAB is known to form a dense bilayer on AuNR due to the strong binding affinity of bromide ions to gold^[R1, 2]. In contrast, CTAC forms a more loosely packed bilayer, thereby allowing surface-initiated reactions to proceed more readily. In addition, the higher reduction potential of PtCl_4^{2-} compared to PtBr_4^{2-} (0.75 V vs. 0.70 V for $\text{PtCl}_4^{2-}/\text{Pt}$ and $\text{PtBr}_4^{2-}/\text{Pt}$, respectively)^[R3, 4] may further facilitate the redox reaction between the platinum and iron precursors.

In response to the reviewer's comment, we have provided more detailed discussions on the use of K_2PtCl_4 and surfactant exchange as follows. We also provided an experimental data showing that the surfactant exchange is necessary for the deposition of FeOOH on AuNRs (Figure S5) in our revised SI.

(Main text, Page 4, Line 23) "Prior to Fe_xO_y coating, the hexadecyltrimethylammonium bromide (CTAB) medium of the AuNR solution was exchanged with a hexadecyltrimethylammonium chloride (CTAC) solution to facilitate further reactions. Then, K_2PtCl_4 and $\text{FeCl}_2 \cdot 4\text{H}_2\text{O}$ were sequentially added to the AuNR solution, and the mixture was heated at 100 °C for 1 h. The redox reaction between PtCl_4^{2-} and Fe^{2+} generates an initial iron oxyhydroxide (FeOOH) coating on AuNRs, upon which further deposition of FeOOH occurs through the oxidation of Fe^{2+} by dissolved oxygen²⁶. Our optimized experimental condition (0.4 mM of K_2PtCl_4 , 10 mM of $\text{FeCl}_2 \cdot 4\text{H}_2\text{O}$, and 4.4 nM of AuNRs) resulted in a uniform FeOOH shell with a thickness of 14 ± 1 nm (Figure 1c; Figure S1). The small amount of K_2PtCl_4 was required to form a uniform FeOOH shell (Figure S2), and the shell thickness and morphology were controlled by adjusting the ratio of AuNRs to iron precursors (Figure S3 and S4). Without the CTAB-to-CTAC surfactant exchange step mentioned above, the FeOOH shell formation was significantly suppressed, as shown by TEM and extinction spectra (Figure S5). CTAC is reported to form a less densely packed molecular layer on AuNRs than CTAB^{27, 28}, allowing surface-initiated reactions to occur more readily. In addition, the higher reduction potential of PtCl_4^{2-} compared to PtBr_4^{2-} (0.75 V and 0.70 V for $\text{PtCl}_4^{2-}/\text{Pt}$ and $\text{PtBr}_4^{2-}/\text{Pt}$, respectively)^{29, 30} may further facilitate the surface-initiated redox reaction between the platinum and iron precursors."

(SI, Page 9, Figure S5)

Figure S5. Extinction spectra (black: as-synthesized AuNR, red: AuNR subjected to FeOOH coating procedure) and TEM images of AuNRs subjected to FeOOH coating procedure under (a) CTAB (i.e., without surfactant change) and (b) CTAC conditions (i-ii: redox reactions involved in FeOOH formation).

Comment 3: After performing annealing to obtain the AuNR/IO/SiO₂ nanoparticles, they were redispersed through sonication in water. Was any sintering between nanoparticles observed, and did they redisperse well?

Response: AuNR/IO/SiO₂ particles were well-dispersed in water with a brief sonication (~5 s), as confirmed by dynamic light scattering (DLS) analysis (Figure S10). In response to the reviewer's comment, we added the DLS data in our revised SI with the following statement. To further support this observation, we have included a supplementary video demonstrating the uniform dispersion after sonication (Response_Movie1).

(Main text, Page 6, Line 2) “The synthesized AuNR/Fe_xO_y/SiO₂ powder was dispersed in water by brief sonication (~5 s). The aqueous dispersion was stable without aggregation, as confirmed by dynamic light scattering (DLS) analysis (Figure S10).”

(SI, Page 2, Line 21) “Dynamic light scattering (DLS) measurements were performed using a Malvern Panlytical Zetasizer Nano-ZS analyzer equipped with a 632.8 nm laser at a scattering angle of 173°.”

(SI, Page 12, Figure S10)

Figure S10. DLS data of MPs after dispersion in water by brief sonication. The measured diameter was 95.7 nm with a polydispersity index (PDI) of 0.188.

Comment 4: The experimental results through crossed polarizers are quite interesting, but the discussion of birefringence should be improved. How can the authors distinguish between birefringence and linear dichroism? This sentence mentions scattering: “If the phase difference between the two scattered components is close to...” Why is scattering emphasized over absorption? Given the small size of the nanorods, I would expect relatively little scattering, and absorption would be the main mechanism of extinction.

Response: We thank the reviewer for the insightful comment regarding the distinction between birefringence and linear dichroism. We agree that it is important to clarify how these two effects differ and how our observations support the presence of birefringence rather than dichroism.

This distinction is indeed central to interpreting polarization-dependent optical responses in anisotropic systems. Although both effects arise from structural or material anisotropy, their manifestations are fundamentally different. Linear dichroism occurs when a material exhibits anisotropic absorption, such that one polarization component is attenuated more strongly than its orthogonal counterpart. This leads to an intensity difference between the two polarization channels but does not alter their phase relationship. Birefringence, on the other hand, causes a phase delay between the two polarization components of the incident light. This phase delay modifies the polarization state itself, transforming linear polarization into elliptical or rotating its orientation—which can lead to measurable transmission even under cross-polarized detection conditions. In our experiments, light transmission was observed under cross-polarized conditions when the sample was oriented at 45° to input polarization. This behavior indicates that the incident polarization state was rotated ~90 degrees during transmission,

which is a direct consequence of birefringence. If linear dichroism were the dominant mechanism, such a dramatic rotation would not be observable, and transmission through the crossed analyzer would be trivial.

Additionally, we appreciate the reviewer's concern regarding the use of the term "scattered components" in our original description. In the context of conventional extinction measurements under single polarization illumination, such as those presented in Figure 2, we agree that absorption is the dominant contributor. This is consistent with the size regime of our nanorods, where the particle dimensions are below 100 nm and radiative scattering is relatively weak compared to absorptive losses. However, in the cross polarized configuration, where light is transmitted through a polarizer, the sample, and an analyzer oriented at 90 degrees, the observed signal originates from the component of the electric field that has undergone polarization rotation. This rotation arises due to a phase difference between the orthogonal polarization components introduced by the anisotropic optical response of the nanorods. As a result, even though the incident light is blocked by the crossed analyzer in the absence of rotation, a portion of the field emerges along the analyzer axis when such phase retardation occurs. The detector records the direct field and the scattered field, but since the direct field is blocked by the crossed analyzer, the recorded signal is the scattered field.

To avoid confusion and to clarify the origin of the observed effect, we revised the wording in the manuscript. Specifically, the term "scattered components" was replaced with "field components" to better reflect the actual mechanism involving phase-shifted output field components that result from plasmon-induced anisotropy. We believe this clarification more accurately conveys that the measured signal arises from polarization rotation due to phase retardation.

(Main text, Page 10, Line 18) "No light transmission was observed under the cross-polarization setting of $\theta_1 = 0^\circ/\theta_2 = 90^\circ$ or $\theta_1 = 90^\circ/\theta_2 = 180^\circ$ (Figure 3a), as expected, since there is no mechanism for the electric field to rotate into alignment with the output polarizer."

(Main text, Page 11, Line 5) "Each component excites its respective SPR mode, accompanied by a phase retardation. If the phase difference between the field components is close to π , the polarization rotates by 90° (Figure 3b, II), allowing light to transmit through the cross-polarizer."

(Main text, Page 13, Line 4) "Consequently, the field mostly consists of the parallel component, which is blocked when the output polarizer is aligned along the perpendicular direction (i.e., $\theta_2 = 90^\circ$)."

(Main text, Page 13, Line 18) "the resultant electric field reorients from 45° to 135° (Scheme S1), enabling strong transmission intensity through the cross-polarizer."

Comment 5: For comparison, what are the highest CD angle and g-factor previously reported for chiral assemblies of achiral plasmonic nanoparticles?

Response: Below, we provide a list of notable g-factors from solution phase assemblies of achiral plasmonic nanoparticles.

1. DNA-based assembly: DNA origami-based assemblies exhibited g-factors in the range of 0.01–0.02 [R5-8].
2. Peptide-based assembly: CD signals around 100 mdeg have been reported for peptide-based assemblies [R9, 10]. A recent study by Cheng et al. achieved an exceptionally high g-factor of 0.12 from a long-range order peptide-based self-assembly [R11].
3. Biological assembly: A helical bacteria-based assembly showed g-factors on the order of 10^{-4} to 10^{-3} [R12].
4. Magnetic assembly: Previous studies on magnetic alignment of plasmonic nanoparticles exhibited g-factors in the range of 0.01~0.04 [R13-15].

Our system achieved a g-factor of 0.21, which, to the best of our knowledge, is the highest reported value among solution phase assemblies of achiral nanoparticles. We have cited the work listed above in our revised manuscript.

Comment 6: More references for magnetically functionalized gold nanorods and their alignment in magnetic fields should be added.

Response: We appreciate the reviewer's suggestion. In the revised manuscript, we have cited additional reports on magnetically functionalized gold nanorod and their magnetic alignment.

(Main text, Page 3, Line 13) "Several different approaches have been reported for synthesizing magnetically controllable plasmonic nanoparticles (MPs)¹³⁻²¹."

(Main text, Page 23, Line 8)

17. Gwak, J., *et al.* Reconfigurable Metasurface of Magnetoplasmonic Microbundle Array for Chiral Signal Enhancing. *Adv. Opt. Mater.* **11**, 2202104 (2022).
20. Wang, M., *et al.* Magnetic tuning of plasmonic excitation of gold nanorods. *J. Am. Chem. Soc.* **135**, 15302-15305 (2013).
21. Rizvi, M. H., *et al.* Magnetic Alignment for Plasmonic Control of Gold Nanorods Coated with Iron Oxide Nanoparticles. *Adv. Mater.* **34**, e2203366 (2022).

Reviewer 2

The authors developed a new synthetic approach for magnetically controllable plasmonic nanoparticles (MPs) composed of an anisotropic gold core and an iron oxide (IO) shell, enabling rapid magnetic alignment control while preserving tunable plasmonic properties. Linearly aligned MPs exhibit reconfigurable birefringence with polarization-dependent color tuning, while helically arranged MPs achieve record-breaking extrinsic chirality (g -factor = 0.21) through magnetic field-induced assembly of superstructures. The result shows broad applicability for smart optical devices requiring real-time plasmonic and chiral modulation. I would recommend the publication of this manuscript on Nature Communications after minor revisions.

Response: We sincerely thank the reviewer for the positive and helpful comments.

Comment 1: Could the authors provide experimental data (e.g., XRD and XPS) obtained under different thermal treatment conditions to clarify the transformation from IOH to IO and confirm the phase composition of the resulting IO?

Response: We carried out additional XRD measurements for a series of samples treated at different thermal conditions under reductive environment (Figure S7), which indicates the transformation from iron oxyhydroxide (β -FeOOH) to iron oxide (Fe_2O_3 and Fe_3O_4) and further to elemental iron with increasing the temperature and reaction time.

(SI, Page 11, Figure S7)

Figure S7. XRD profiles of (a) AuNR/FeOOH/SiO₂, (b) AuNR/Fe_xO_y/SiO₂ (reduction condition of 310 °C and 2 h). The XRD patterns of AuNR/FeOOH/SiO₂ and AuNR/Fe_xO_y/SiO₂ were dominated by the strong gold signals, making it difficult to analyze

the porous iron-containing layer. (c) XRD profiles of FeOOH (before reduction) and its reduction products formed at different conditions (temperature and duration). Black lines are literature data from the Crystallography Open Database (COD).

It is difficult to distinguish Fe₂O₃ and Fe₃O₄ by XRD and therefore XPS measurements were carried out for two iron oxide samples reduced at 260 °C and 310 °C (Figure S8b-c). The deconvolution of Fe 2p band indicated the existence of both Fe₂O₃ and Fe₃O₄ in both samples with increasing Fe²⁺/Fe³⁺ and Fe₃O₄/Fe₂O₃ ratios with increasing temperature (Table R1).

(SI, Page 11, Figure S8)

Figure S8. (a) XPS spectra of Fe_xO_y prepared by reducing FeOOH at 310 °C for 2 h, showing the presence of Fe 2p, O 1s, C 1s, and Fe 3p peaks with binding energies at 711, 530, 284, and 56 eV, respectively. (b-c) The deconvoluted Fe 2p spectra of Fe_xO_y prepared by reducing FeOOH at (b) 310 °C for 2 h or (c) 260 °C for 30 min displays peaks at 711 eV (Fe 2p_{3/2}) and 724 eV (Fe 2p_{1/2}). Fe 2p spectra were deconvoluted into characteristic peaks, with fitted positions at 711.3 eV (Fe³⁺ 2p_{3/2}), 710.2 eV (Fe²⁺ 2p_{3/2}), 724.8 eV (Fe³⁺ 2p_{1/2}), and 723.5 eV (Fe²⁺ 2p_{1/2}).

Table R1. Estimated Fe²⁺/Fe³⁺ and Fe₃O₄/Fe₂O₃ ratios derived from peak fitting of the Fe 2p XPS spectra for Fe_xO_y annealed under different conditions.

	310 °C, 2 h	260 °C, 30 min
Fe ²⁺	28%	14%
Fe ³⁺	72%	86%
Fe ₃ O ₄	77%	33%
Fe ₂ O ₃	23%	67%

In response to the reviewer's comment, we updated XRD and XPS results in SI with the additional data (Figure S7-S8).

Comment 2: Could the author explain the origin of the chiral optical response observed in isotropic gold nanospheres under helical magnetic fields?

Response: We attribute the weak but measurable chiroptical activity of AuNS/Fe_xO_y/SiO₂ to the formation of linear assemblies of nanoparticles through magnetic dipole interactions under the magnetic field. [R16, 17] The imperfection in the spherical shape of our particles can also contribute to the weak but measurable chiroptical property.

In response to the reviewer's comment, we added the following statement in the revised manuscript.

(Main text, Page 17, Line 1) "Isotropic AuNS/Fe_xO_y/SiO₂ also displayed CD signals under a helical magnetic field, presumably due to the imperfection in the spherical shape and the formation of linear assemblies of NPs."

Comment 3: Could the authors explain why, as the aspect ratio of the nanorod structure increases from 2.3 to 3.4, the circular dichroism of the helically arranged MPs becomes larger, while the g-factor tends to decrease or remain unchanged (Fig. 5c)?

Response: The g-factor is defined as $g = \Delta A / A_{\text{avg}}$, where ΔA represents the difference in absorption between left- and right-circularly polarized light and A_{avg} is the average absorbance of left- and right-circularly polarized light. Therefore, g-factor provides a measure that compensates for the variations in the absorbance (i.e., concentration, path length, extinction coefficient, etc), whereas the CD signal varies with the absorbance. Accordingly, g-factor may remain unchanged or decrease slightly even when the CD signal increases.

Comment 4: Are there any other methods to measure the orientation of MPs except by directly calculating the direction of the magnetic field? Does the measurement result of the chiral optical response depend on the illumination direction in the authors' experiments?

Response: To visualize the orientation of MP assemblies, we conducted dark-field optical microscope imaging at three different locations along the light path (Figure S29): the entry point, midplane, and exit point of the sample. The optical microscope images indicate that MPs form microscale anisotropic assemblies under the magnetic field, which aligns with the magnetic field. The MP assemblies appeared shorter at the entry and exit points (indicating tilted orientation), and longer in the center (indicating horizontal orientation), consistent with the simulated helical field map (Figure 5b-c). In response to the reviewer's comment, we added the new data in the revised Supporting Information with the corresponding statement in the main text.

(Main text, Page 16, Line 16) "The helical arrangement of MPs was supported by dark-field optical microscopy (Figure S29)."

(SI, Page 22, Figure S29)

Figure S29. Reflection-mode optical microscopy analysis of MP alignment under a helical magnetic field. (a) Schematic of the optical path and three sampling positions across the cuvette (① Entry, ② Midplane, ③ Exit). (b–d) Reflection-mode dark-field optical images of MP assemblies fixed in a hydrogel at each position. The projected lengths of the assemblies are shortest at the entry (b) and exit (c), and longest at the midplane (d), indicating a progressive axial tilting of the particle chains along the light propagation direction.

(SI, Page 1, Line 23) “poly(ethylene glycol) diacrylate (PEGDA, average M_n 575), N,N'-methylenebisacrylamide ($\geq 99.5\%$), and glycerol (ReagentPlus®, $\geq 99.0\%$) were purchased from Sigma-Aldrich (St. Louis, MO, USA).”

(SI, Page 1, Line 29) “Ammonium persulfate was purchased from Bio-Rad (Hercules, CA, USA).”

(SI, Page 2, Line 19) “Dark field images were captured using an Olympus BX53M microscope equipped with a halogen lamp, a dark-field condenser, and a 50× MPlanFL N objective lens ($NA = 0.80$).”

(SI, Page 7, Line 3) “**Hydrogel fixation of AuNR/ Fe_xO_y /SiO₂ alignment under a helical magnetic field.** To preserve and visualize the helical arrangement of magnetically controllable plasmonic nanoparticles (MPs), the particles were immobilized in a polyacrylamide-based hydrogel. Specifically, 0.2 g of acrylamide, 0.19 mL of acrylic acid, and 25 mg of N,N'-methylene bisacrylamide were dissolved in 0.125 mL of water with

sonication. Subsequently, 0.875 mL of glycerol was added to the solution and the mixture was sonicated for 10 s. Then, 20 mg of ammonium persulfate, acting as an initiator for polymerization, was added to 1 mL of the monomer mixture. MPs were then dispersed in the solution, referred to as the pre-gel solution, at a concentration of 1 mg/mL. To cast the gel, the MP-containing pre-gel solution was injected between two glass slides with a ~1 mm PDMS spacer. During the gelation, a helical magnetic field was applied by placing two bar magnets in the cross configuration described above. The gelation was initiated by exposing the pre-gel solution to UV light (365 nm) for 2 minutes, which fixed the magnetic alignment of MPs within the hydrogel.”

To address the reviewer’s comment on the illumination direction, we performed CD measurements while reversing the magnetic field orientation relative to the incident light. The resulting spectra showed no significant difference between the two configurations. This result indicates that the observed CD spectra reflect the inherent chirality of MP assemblies under a helical magnetic field.

Figure R1. CD of helically assembled MPs measured under reversed illumination configurations.

Reviewer 3

The controlled production of nanoparticles with reconfigurable optical activity is certainly a topic of great interest for a broad audience, which makes this manuscript potentially adequate for the Nature Communications readership after some improvements are done. While the synthetic approach and the post-synthetic treatment of the different plasmonic nanoparticles are described in detail, the protocols, both for experiments and simulations, are too succinct to allow the replication of the results and the lacking information also makes it difficult to understand the underlying physics.

Response: We thank the reviewer for the positive and helpful comments. In our revised manuscript, we have provided more detailed descriptions on the experimental and simulation procedures to ensure that the work can be reproduced by others and to help readers better understand the underlying physics. We also provided additional explanations and references on the underlying principles as follows.

(Main text, Page 13, Line 13) “Since the incident electric field is composed of two orthogonal components, a phase shift in either component leads to a progression of polarization states—from linear to elliptical, to circular, and back to linear—with the final linear polarization exhibiting a different orientation (Scheme S1). The phase difference is known to transition from 0 to $\sim\pi$ across the plasmonic resonance wavelength, amounting to $\pi/2$ at the resonance peak³⁸.”

(SI, Page 18, Scheme S1)

Scheme S1. Schematic illustration of polarization rotation induced by phase shift. The incident electric field (yellow curve) is composed of two orthogonal components (black curves). Their superposition gives rise to elliptical, circular, or linear polarization states, depending on the relative phase shift— $\pi/4$, $\pi/2$, and π , respectively. At a phase shift of π , the resulting polarization returns to a linear state, rotated by 90° with respect to the original polarization direction.

Comment 1: For instance, the electrodynamic simulations contain no description at all of the size and shape of the model nanoparticles, the authors only say what software has been used and the dielectric constant of each material, which would not be enough if I wanted to replicate their FDTD simulations.

Response: We thank the reviewer for the helpful comment. We performed simulations for six representative AuNR, each featuring different AuNR dimensions, reflecting the structural variation observed in the synthesized nanostructures. The geometries were initially based on TEM measurements, but we allowed slight adjustments within the estimated experimental uncertainty to better match the extinction features observed in the optical measurements. This approach was necessary to ensure meaningful comparison between the simulations and the measured spectra, while still remaining faithful to the physical structure of the system.

To support full reproducibility, we now provide a complete summary of the simulation geometries and parameters in Supplementary Table S1. We also expanded the description of the simulation settings to include the polarization configurations, wavelength range, mesh resolution, and boundary conditions. For example, we now clearly state that a 0.5 nm conformal mesh was applied around the nanostructure, and perfectly matched layers (PML) were used in all directions to eliminate boundary reflections. These additions should enable others to fully reconstruct and validate our simulations.

(SI, Page 5, Line 18) “**Finite-Different Time-Domain (FDTD) Modeling.** Numerical calculations were performed using commercial FDTD software (Lumerical FDTD, Lumerical Solutions) to calculate the extinction cross-section of AuNS/Fe_xO_y/SiO₂ and AuNR/Fe_xO_y/SiO₂ nanostructures. The simulations were performed on a AuNR/Fe_xO_y/SiO₂ and six representative AuNR/Fe_xO_y/SiO₂ with different NP geometries, as summarized in Table S1. Each sample consisted of a AuNS or AuNR with specific dimensions and aspect ratio, coated with Fe_xO_y and SiO₂ shells of varying thicknesses. The modeled structures reflect experimentally synthesized NPs whose dimensions were measured by TEM analysis. In some cases, the NP dimensions were slightly adjusted within the TEM-estimated error range to better match the experimentally observed extinction peaks. The optical constants were obtained from previously reported data, using values from Johnson and Christy⁸ for gold, from Palik⁹ for SiO₂, and a constant refractive index of 2.3 for Fe_xO_y across the relevant wavelength range¹⁰. The surrounding medium was set to water with a refractive index of 1.33. A total-field scattered-field (TFSF) source was applied over the 480 to 900 nm wavelength range. To capture the polarization-dependent plasmonic response, two excitation conditions were used, one with the electric field aligned parallel to the nanorod’s long axis to excite the longitudinal surface plasmon resonance (P_{\parallel}), and the other with the field perpendicular to it to excite the transverse mode (P_{\perp}). A conformal mesh with a minimum resolution of 0.5 nm was applied near the nanostructures to ensure sufficient spatial accuracy.

The simulation domain was enclosed by perfectly matched layers in all directions to eliminate artificial reflections at the boundaries. The extinction cross-sections were obtained by integrating the absorbed and scattered power throughout the simulation region.”

(SI, Page 6, Table S1)

Table S1. Geometrical parameters of AuNS/Fe_xO_y/SiO₂ (1) and AuNR/Fe_xO_y/SiO₂ (2-7) structures used in FDTD simulations. Values were based on TEM analysis with minor adjustments for optical matching.

Sample No.	Width of AuNR (nm)	Length of AuNR (nm)	Aspect ratio	Thickness of Fe _x O _y (nm)	Thickness of SiO ₂ (nm)
1	40	40	1.0	3.0	20
2	40	60	1.5	3.0	15
3	18	32	1.8	3.0	20
4	35	77	2.2	3.0	25
5	30	69	2.3	3.0	22
6	30	81	2.7	2.8	25
7	18	61	3.4	3.0	25

Comment 2: But what I missed the most was a more clear and detailed description of how the magnetic field was applied to the samples in each experiment, which is the core of this investigation. Beginning with the TEM characterization depicted in Fig. S10, the illustration has no description of the experimental setup, authors should indicate clearly what each graphical element stands for.

Response: We thank the reviewer for the helpful comment. In response, we have updated the figure (Figure S12 in the revised manuscript) to label each element in the graphic as well as red arrows for the magnetic field direction. Additionally, we have revised the figure caption to provide a more detailed description of our experimental set-up.

(SI, Page 14, Figure S12)

Figure S12. (a) Schematic illustration of the experimental setup for linear magnetic field generation for TEM sampling. TEM images of MPs (b) with and (c) without an external magnetic field (18 mT). A droplet of MP solution was placed on a TEM grid between a pair of cylindrical NdFeB magnets (3.8 cm diameter \times 1.9 cm thick, grade N52, K & J Magnetics, Inc.) separated by a plastic rack (end-to-end distance of 18 cm), which was then left overnight for drying. The upper edge of the TEM grid was cut along the direction of the applied magnetic field to mark the field direction.

Comment 3: Scheme S1 has a more complete description of the elements comprising the experimental setup, but it does not indicate where the magnets are. I might assume the magnet is the gray block on the left of the sample, but if this is the case then the magnetic field is not homogeneous across the samples, as would be the case for the pair of parallel planar magnets depicted in Fig. S10 (assuming that the red/blue cylinders are the magnets).

Response: The reviewer is correct that the gray block is the magnet. We have updated Scheme S2 to label each component of the graphic including the magnet.

(SI, Page 19, Scheme S2)

Scheme S2. Schematic illustration of the setup for optical imaging of MP solutions under a magnetic field and input/output polarizers. A bar-shaped NdFeB magnet (gray block) is positioned adjacent to a vial containing the MP solution to apply the linear magnetic field (18 mT). The red arrow indicates the magnetic field direction. The inset (red box) presents the magnetic field distribution calculated by Ansys Maxwell Electromechanical Device Analysis Software, confirming the linear magnetic field at the sample position (inside the blue circle). The input (θ_1) and output (θ_2) polarizer angles are defined relative to the applied magnetic field direction (blue dotted line).

We understand the reviewer's concern on the uniformity of the magnetic field. We have examined the magnetic field distribution in our experimental setup by Ansys simulation (Scheme S2, inset). Specifically, we modeled a bar-shaped NdFeB magnet ($50 \times 20 \times 10$ mm, grade N35) placed adjacent to the sample vial, with air as the surrounding medium. The simulated field vectors, evaluated in a plane perpendicular to the optical axis, indicated a uniform field in the measurement region (inside the blue circle in the inset (red box) of Scheme S2). While more uniform fields may be generated using symmetrically arranged magnets, we chose to use a single magnet configuration for safety, as strong neodymium magnets can collide when placed in close proximity.

The field uniformity is further supported by the uniform color distribution observed in the transmission image below (Figure R2a); substantial inhomogeneity in the magnetic field results in noticeable color variations as shown in Figure R2b.

Figure R2. Transmission images of MP solutions between crossed polarizers under (a) uniform, and (b) non-uniform magnetic field.

Comment 4: The same issues apply to Fig. 5 and Fig. S27, which mention the alignment of the magnets with respect to the direction of light, but the cartoon misses proper description and the text is also unclear about what these angles mean. It is also unclear what are the magnetic dipoles depicted in Figs. 5b and S27 - are they the magnetic dipoles of each nanoparticle? or are they projections of the magnetic field between the magnets? In either case, this takes me back to the description of the simulations of the magnetic fields, which is also too succinct to allow replication, the SI text only says what software was used and the dimensions of the magnets.

Response: We thank the reviewer for the helpful comment. The 45° and 135° angles in Figure 5a represent the angles between the light propagation direction (dashed line, y-direction) and the orientation of bar magnets (gray bar). Two bar magnets placed at 45° induces the left-handed helical field. Two bar magnets placed at 135° induces the right-handed helical field. In response to the reviewer's comment, we added more explicit descriptions in the revised figure caption.

The arrows in original Figure 5b and Figure S27 represented the simulated magnetic field distribution taken every 1 mm along the light propagation direction (y-axis), generated by the magnet configuration described in Figure 5a. In the revised manuscript, we replotted the magnetic field simulation data from a different viewing angle and updated the figure caption to enhance readability.

(Main text, Page 18, Figure 5)

Figure 5. Extrinsic chirality of helically arranged MPs. (a) Experimental set-ups used to induce left-handed (left side) or right-handed (right side) helical magnetic fields. Two bar magnets (gray flat rectangular bar) were positioned orthogonally in a cross configuration, with their axes oriented at 45° (left) and 135° (right) relative to the light propagation direction (yellow arrow, y axis). (b) Schematic illustration of the cross-sectional planes used in the magnetic field simulations. Each color represents a specific y -position along the light propagation direction (red: -0.5 cm; orange: -0.25 cm; green: 0 cm; blue: $+0.25$ cm; purple: $+0.5$ cm). The origin $(0, 0, 0)$ is defined as the midpoint of the line connecting the centers of the two bar magnets. (c) Vector plots of simulated magnetic fields for the cross

configurations of bar magnets (NdFeB magnet, $20 \times 10 \times 1$ mm) placed with an end to-end separation distance of 1.5 cm. The arrows represent the magnetic field direction marked every 0.25 cm along the light propagation direction (y direction). The color of each vector corresponds to a specific y-position marked in b. The black dot represents the central point of the simulation plane at $(x, z) = (0, 0)$. The red and blue curved arrows indicate the rotational orientation of the magnetic field with respect to the light propagation direction. The variations in the apparent lengths of the magnetic field vectors arise from their tilt relative to the x–z plane. The scale bars represent 200 μm for both plots. (d) CD and g-factor spectra of MPs with AuNR aspect ratios of 2.3, 2.7, 3.0, and 3.4 under helical magnetic fields (20 mT) with 45° (red) and 135° (blue) magnet settings along with the data collected without a magnetic field (black).

We also added the following description for clarity and provided more detailed description on the magnetic field simulation method in SI as follows.

(Main text, Page 16, Line 9) “This cross configuration of bar magnets generates a helical distribution of magnetic field across the sample, confirmed by the magnetic field simulation (Figure 5b and c) where the magnetic field vectors show gradual rotation along the light propagation direction.”

(SI, Page 6, Line 17) “**Simulation of the magnetic field.** Ansys Maxwell Electromechanical Device Analysis Software was used to simulate the magnetic field distribution for two different magnet settings used to induce left-handed and right-handed helical magnetic fields, in which two bar magnets (NdFeB magnet, $20 \times 10 \times 1$ mm) were positioned orthogonally in a cross configuration, with their axes oriented at 45° (left-handed helical field) or 135° (right-handed helical field) relative to the light propagation direction, and placed with an end-to-end separation of 1.5 cm (Figure 5a). NdFeB-N35 was used as the magnet material, and air was chosen as the ambient condition. To visualize the spatial distribution of the magnetic field, field distribution was extracted on cross-sectional planes perpendicular to the light propagation direction (optical axis) at 0.25 cm intervals. These slices were subsequently overlaid to reconstruct a three-dimensional field representation, as shown in Figures 5c.”

Comment 5: At this point, I would not analyze the results in depth because of the many doubts I have about the experimental and computational setups, so I request that authors should deepen the description of each and every experiment and simulation. If I may suggest, authors should include in all cartoons vectors depicting the direction of the magnetic field, since this is the origin of the reconfigurable optical activity being discussed all along the manuscript. These details matter for the broader audience that awaits ahead, since this broader audience is much less familiar with the topics being discussed than the expert reviewers handling the manuscript at this stage.

Response: We thank the reviewer for the constructive suggestion. We depicted field directions in all cartoons and carefully revised the manuscript to provide detailed descriptions of the experimental and simulation methods for a broad audience. We hope the reviewer finds the revised manuscript satisfactory.

Reviewer 4

The manuscript authored by Kang delineates the synthesis and dynamic (chir)optical properties of magneto-plasmonic nanorods (MP) manipulated by magnetic field. The MPs synthesized by the growth of iron oxide on the surface of Au nanorods show excellent superparamagnetic properties, which can be aligned by an external magnetic field. Owing to the dual resonance modes inherent to the gold nanorods, the aligned MPs exhibit birefringence property under unidirectional linearly polarized light. More interestingly, MPs can be aligned by helical magnetic field to produce chiral assembly of MPs with outstanding chiroptical properties with a g-factor up to 0.21. I appreciate the high innovation of this work and the clarity of the data analysis. I would suggest this work for publication only after a minor revision.

Response: We sincerely thank the reviewer for the positive and helpful comments.

Comment 1: The authors claimed in the manuscript that this work has achieved the highest g-factor (0.21) for dynamic nanoparticle assemblies to date. However, to the best of my knowledge, a recent work on the chiral assembly of chiral gold nanorods has reported a g-factor of 0.24 (Nano Lett. 2024, 24, 41, 13027-13036). Therefore, the authors need to revise their statement, although it would not affect the impact of the work.

Response: We thank the reviewer for the helpful comment. The recent study^[R18] referred by the reviewer achieved a g-factor of 0.24 from the assembly of chiral nanorods. We have cited the paper as ref 46 in our revised manuscript. We also revised our statement to clarify that the g-factor of 0.21 in our system represents the highest reported value for solution phase assemblies of achiral nanoparticles.

(Main text. Page 1, Line 25) “the highest reported value for **solution-phase assemblies of achiral nanoparticle.**”

(Main text. Page 4, Line 12) “Furthermore, the MPs under a helical magnetic field demonstrated giant chirality with a g-factor of 0.21, **which is among the highest values reported thus far for plasmonic assemblies.**”

(Main text. Page 16, Line 21) “The g-factor observed in this work is among the highest reported **values for plasmonic systems⁴¹⁻⁴⁴ and constitutes the highest value reported thus far for dynamic assemblies of achiral NPs^{, 13, 14, 16, 17, 25, 40, 45-54}**”

(Main text, Page 25, Line 8)

46. Zhang, N. N., *et al.* Self-Matching Assembly of Chiral Gold Nanoparticles Leads to High Optical Asymmetry and Sensitive Detection of Adenosine Triphosphate. *Nano Lett.*, 13027–13036 (2024).

Comment 2: Please indicate the direction of the magnetic field in Fig. 3 and Fig. 4, as those shown in Fig. 2, and emphasize the distinction between the magnetic field and the optical field in the figures.

Response: We thank the reviewer for the helpful suggestion. In response, we marked the applied magnetic field in each scheme of Figure 3 and 4 with red arrows. We color-coded the electric and magnetic fields using black and red lines respectively for further clarity and indicated that in the Figure captions.

(Main text. Page 9, Figure 2)

Figure 2. Optical properties of MPs under B_x . (a) Schematics depicting plasmonic excitation under unpolarized light, $P_{||}$ and P_{\perp} . Black and red arrows indicate the direction of light polarization and applied magnetic field, respectively. (b-d) Photographs and extinction spectra (measured: solid lines; simulated: dotted lines) of MPs incorporating AuNRs with an aspect ratio of 2.3 under unpolarized light (b), $P_{||}$ (c), and P_{\perp} (d). The experimental extinction spectra appear broader than the simulated spectra due to ensemble averaging over MPs with a

size distribution. (e-g) Extinction spectra and photographs of MPs containing AuNRs with varying aspect ratios under unpolarized light (e), P_{\parallel} (f), and P_{\perp} (g). The applied magnetic field strength was measured to be 18 mT.

(Main text. Page 12, Figure 3)

Figure 3. Birefringent colors from linear MP arrays formed under B_x . (a) Schematic illustration ($\theta_1 = 90^\circ/\theta_2 = 180^\circ$) and photographs showing no light transmission at the cross-polarization condition of $\theta_1 = 90^\circ/\theta_2 = 180^\circ$ or $\theta_1 = 0^\circ/\theta_2 = 90^\circ$, where θ_1 and θ_2 denote input and output polarizer angles relative to the magnetic field direction (red arrow, x-axis). (b) Schematic illustration ($\theta_1 = 45^\circ/\theta_2 = 135^\circ$) and photographs showing light transmission at the cross-polarization condition of $\theta_1 = 45^\circ/\theta_2 = 135^\circ$ or $\theta_1 = 135^\circ/\theta_2 = 45^\circ$. Flat gray arrows indicate the axes of polarizing filters. Yellow sinusoidal lines represent the electric field waves of the incident light. A π phase shift between two orthogonal field components (dashed yellow lines in b) from region I to II results in a 90° rotation of the polarization direction. MPs containing AuNRs with an aspect ratio of 2.3 were used for the experiments. Photographs were taken under the applied magnetic field strength of 18 mT and white light illumination.

(Main text. Page 14, Figure 4)

Figure 4. Plasmon-mediated birefringent responses from linearly aligned MPs under varying θ_2 at a fixed θ_1 of 45°. (a) Schematics illustrating light transmission at $\theta_2 = 90^\circ$, 135° , and 180° at T-SPR, near L-SPR, and L-SPR wavelength, respectively. Red arrows indicate the direction of the applied magnetic field (x-axis). Flat black arrows indicate the axes of polarizing filters, where θ_1 and θ_2 denote polarizer angles relative to the magnetic field direction (dotted lines). Yellow sinusoidal lines represent the electric field wave of the incident light. (b-c) photographs (b) and spectra (c) of light transmission from linear MP (AuNR aspect ratio: 2.3) arrays at θ_2 of 90°, 135°, or 180°. (d) Photographs of transmitted colors from linear MP arrays with varying AuNR aspect ratios under various θ_2 . (e) Transmission spectra of linear MP arrays with AuNR aspect ratios of 1.9 (top), 2.3 (middle), and 3.4 (bottom) collected at θ_2 of 90°–180°. All experiments were conducted under the applied magnetic field strength of 18 mT and θ_1 of 45°.

Comment 3: What was the strength of the applied magnetic fields?

Response: The magnetic field strength was measured to be 18 mT for linear alignment and 20 mT for helical alignment, as stated in our originally submitted SI. We have indicated the field strength for each measurement in the Figure caption of our revised manuscript and added a following sentence in the method section of SI.

(SI, Page 2, Line 18) “Magnetic field strength was measured using a 455 DSP Gauss meter.”

Comment 4: I really would like to recommend the authors to use Fe_xO_y to replace IO, which is hardly be used in the field of chemistry or materials.

Response: As requested, we have replaced “IO” with “ Fe_xO_y ” in our revised manuscript and SI. We also replaced IOH with FeOOH for consistency.

Comment 5: For the helical magnetic field alignment, I am wondering what were the extinction spectra of the MP solution? Were the nanorods aligned in a helical way as shown in the cartoon of Figure 5b? If so, what was the pitch distance and interrod distance?

Response: As presented below, the extinction spectra did not change significantly with the application of the helical field under the unpolarized light.

Figure R3. Extinction spectra of MPs (AuNR aspect ratio: 3.0) under helical magnetic fields.

To visualize the orientation of MP assemblies, we conducted dark-field optical microscope imaging at three different locations along the light path (Figure S23): the entry point, midplane, and exit point of the sample. The optical microscope images indicate that MPs form microscale anisotropic assemblies under the magnetic field, which aligns with the magnetic field. The MP assemblies appeared shorter at the entry and exit points (indicating

tilted orientation), and longer in the center (indicating horizontal orientation), consistent with the simulated helical field map (Figure 5b-c). The helical pitch estimated from the simulated magnetic field distribution was approximately 5.6 cm. The inter-rod distance within the subunits is about 50 nm, based on the core-shell dimensions of the MPs.

(SI, Page 22 Figure S29)

Figure S29. Reflection-mode optical microscopy analysis of MP alignment under a helical magnetic field. (a) Schematic of the optical path and three sampling positions across the cuvette (① Entry, ② Midplane, ③ Exit). (b–d) Reflection-mode dark-field optical images of MP assemblies fixed in a hydrogel at each position. The projected lengths of the assemblies are shortest at the entry (b) and exit (c), and longest at the midplane (d), indicating a progressive axial tilting of the particle chains along the light propagation direction.

Other changes

1. Minor changes have been made to correct grammatical and stylistic errors.

REFERENCES

- R1. Mosquera, J., Wang, D., Bals, S. & Liz-Marzan, L. M. Surfactant Layers on Gold Nanorods. *Acc. Chem. Res.* **56**, 1204-1212 (2023).
- R2. Lee, J. H., Gibson, K. J., Chen, G. & Weizmann, Y. Bipyramid-templated synthesis of monodisperse anisotropic gold nanocrystals. *Nat. Commun.* **6**, 7571 (2015).
- R3. Lide, D. R., Ed. *CRC Handbook of Chemistry and Physics, Internet Version 2005*. (CRC Press, Boca Raton, FL, 2005).
- R4. Zhang, H., *et al.* Synthesis of Pd-Pt bimetallic nanocrystals with a concave structure through a bromide-induced galvanic replacement reaction. *J. Am. Chem. Soc.* **133**, 6078-6089 (2011).
- R5. Kuzyk, A., *et al.* DNA-based self-assembly of chiral plasmonic nanostructures with tailored optical response. *Nature* **483**, 311-314 (2012).
- R6. Martens, K., *et al.* Long- and short-ranged chiral interactions in DNA-assembled plasmonic chains. *Nat. Commun.* **12**, 2025 (2021).
- R7. Song, Z., Zhou, Y., Dong, J. & Wang, Q. DNA Origami Guided Helical Assembly of Gold Nanorods with Tailored Optical Chirality. *ACS Appl. Opt. Mater.* **3**, 530-536 (2024).
- R8. Yan, W., *et al.* Self-assembly of chiral nanoparticle pyramids with strong R/S optical activity. *J. Am. Chem. Soc.* **134**, 15114-15121 (2012).
- R9. Song, C., *et al.* Tailorable plasmonic circular dichroism properties of helical nanoparticle superstructures. *Nano Lett.* **13**, 3256-3261 (2013).
- R10. Merg, A. D., *et al.* Peptide-Directed Assembly of Single-Helical Gold Nanoparticle Superstructures Exhibiting Intense Chiroptical Activity. *J. Am. Chem. Soc.* **138**, 13655-13663 (2016).
- R11. Lu, J., *et al.* Enhanced optical asymmetry in supramolecular chiroplasmonic assemblies with long-range order. *Science* **371**, 1368-1374 (2021).
- R12. Feng, W., *et al.* Plasmonic nanoparticles assemblies templated by helical bacteria and resulting optical activity. *Chirality* **32**, 899-906 (2020).
- R13. Li, Z., Fan, Q., Ye, Z., Wu, C., Wang, Z. & Yin, Y. A magnetic assembly approach to chiral superstructures. *Science* **380**, 1384-1390 (2023).
- R14. Wu, C., Fan, Q., Li, Z., Ye, Z. & Yin, Y. Magnetic assembly of plasmonic chiral superstructures with dynamic chiroptical responses. *Mater. Horiz.* **11**, 680-687 (2024).

- R15. Jeong, K. J., *et al.* Helical Magnetic Field-Induced Real-Time Plasmonic Chirality Modulation. *ACS Nano* **14**, 7152-7160 (2020).
- R16. Hu, Y., He, L. & Yin, Y. Magnetically responsive photonic nanochains. *Angew. Chem. Int. Ed. Engl.* **50**, 3747-3750 (2011).
- R17. Song, Y., Tran, V. T. & Lee, J. Tuning Plasmon Resonance in Magnetoplasmonic Nanochains by Controlling Polarization and Interparticle Distance for Simple Preparation of Optical Filters. *ACS Appl. Mater. Interfaces* **9**, 24433-24439 (2017).
- R18. Zhang, N. N., *et al.* Self-Matching Assembly of Chiral Gold Nanoparticles Leads to High Optical Asymmetry and Sensitive Detection of Adenosine Triphosphate. *Nano Lett.*, 13027–13036 (2024).

Response to reviewers' comments

We thank the reviewers and the editor for their positive evaluation and helpful comments. In this final revision, we have addressed the remaining editorial and reviewer comments to ensure full clarity and compliance with the journal's requirements. We truly appreciate the thoughtful comments and suggestions that have helped improve the manuscript throughout the review process, and hope that the editor now finds it suitable for publication.

Reviewer 1

The authors have comprehensively addressed the reviewers' comments and appropriately revised the manuscript. I recommend it for publication after addressing the following interrelated comments. Can the MPs be trapped if a permanent magnet is placed next to a vial containing a dispersion of them, i.e., do they exhibit some level of magnetophoresis? It would be good to mention that for comparison with related magnetoplasmonic nanoparticles. That might also affect the extent to which a magnetic field gradient in Scheme S2 is negligible or not. Does the vector field from the simulations in Scheme S2 indicate the magnitude of the magnetic field in addition to the direction? It could be helpful to calculate the strength of the magnetic field toward the left-center and right-center of those simulations to confirm that the gradient is weak. Why is the simulated field asymmetrical? On the right side, some of the arrows tilt up (characteristic with the field of a magnetic dipole), but none tilt down (which would also be expected if the simulations are performed along a central axis).

Response: We thank the reviewer for the helpful comments. We carried out magnetophoresis measurements using the set-up described in Figure S2 (Supplementary Fig. 24 in the current version) by monitoring birefringent color over time. In this set-up, the transmitted light intensity is expected to decrease with the migration and accumulation of MPs towards the magnet. As shown in Figure R1, slight color change began to occur at around 2 hours and significant darkening was observed at around 6 hours with a clear optical contrast between the left and right sides of the vial.

Figure R1. Photographs showing the magnetophoretic behavior of MPs under the experimental set-up described in Supplementary Fig. 24.

In conclusion, our MPs show magnetophoretic behavior and can be trapped by using a magnet. Nonetheless, the migration speed is slower compared to other magneto-plasmonic systems reported in the literature^[R1-2], where the particle accumulation is complete within 15-30 min. Note that the magnetophoretic property can be controlled by varying the FeOOH coating thickness and reduction condition in our synthesis.

In response to the reviewer's comment on the magnetic field strength, we updated the magnetic field simulation data (Scheme S2 in the previous version, Supplementary Fig. 24 in the current version), using color-coded vectors to represent the local magnetic field strength as well as the field direction (Figure R2). The simulation data indicates that the field gradient across the sample is not substantial, consistent with the experimentally observed slow magnetophoretic movement shown in Figure R1.

The small asymmetry in the original simulation was an artifact caused by the domain boundaries used in the original simulation. To minimize such artifacts, we carried out new simulations using larger simulation domains, which exhibited more symmetric distribution of the magnetic field (Figure R2). We have updated Supplementary Fig. 24 with the new simulation data.

Supplementary Fig. 24. Optical set-up for imaging birefringent colors under a magnetic field. Schematic illustration of the setup for optical imaging of MP solutions under a magnetic field and input/output polarizers. A bar-shaped NdFeB magnet (gray block) is positioned adjacent to a vial containing the MP solution to apply the linear magnetic field (18 mT). The red arrow indicates the magnetic field direction. The inset (red box) presents the magnetic field distribution calculated by Ansys Maxwell Electromechanical Device Analysis Software, confirming the linear magnetic field at the sample position (inside the blue circle).

The magnetic field intensity is represented by the color scale bar. The input (θ_1) and output (θ_2) polarizer angles are defined relative to the applied magnetic field direction (blue dotted line).

REFERENCES

R1. Kim, H., Lee, J. & Park, S. Reconfigurable plasmonic nanostructures for tunable optical responses. *Adv. Mater.* **34**, 2203366 (2022).

R2. Ohulchansky, T. Y. et al. Phospholipid micelle-based magneto-plasmonic nanoformulation for magnetic field-directed, imaging-guided photo-induced cancer therapy. *Nanomed.: Nanotechnol. Biol. Med.* **9**, 1192–1202 (2013).